# FROM EMERGENCE TO CONTROL: PROBING AND MODULATING SELF-REFLECTION IN LANGUAGE MODELS

## ABSTRACT

Self-reflection—the ability of a large language model (LLM) to revisit, evaluate, and revise its own reasoning—has recently emerged as a powerful behavior enabled by reinforcement learning with verifiable rewards (RLVR). While self-reflection correlates with improved reasoning accuracy, its origin and underlying mechanisms remain poorly understood. In this work, *we first show that self-reflection is not exclusive to RLVR fine-tuned models: it already emerges, albeit rarely, in pretrained models*. To probe this latent ability, we introduce Reflection-Inducing Probing, a method that injects reflection-triggering reasoning traces from fine-tuned models into pretrained models. This intervention raises self-reflection frequency of Qwen2.5 from 0.6% to 18.6%, revealing a hidden capacity for reflection. Moreover, our analysis of internal representations shows that both pretrained and fine-tuned models maintain hidden states that distinctly separate self-reflective from non-reflective contexts. Leveraging this observation, *we then construct a self-reflection vector, a direction in activation space associated with self-reflective reasoning*. By manipulating this vector, we enable bidirectional control over the self-reflective behavior for both pretrained and fine-tuned models. Experiments across multiple reasoning benchmarks show that enhancing these vectors improves reasoning performance by up to 12%, while suppressing them reduces computational cost, providing a flexible mechanism to navigate the trade-off between reasoning quality and efficiency without requiring additional training. Our findings further our understanding of self-reflection and support a growing body of work showing that understanding model internals can enable precise behavioral control.

## 1 INTRODUCTION

Reinforcement Learning with Verifiable Rewards (RLVR) has emerged as a powerful technique for enhancing the reasoning abilities of large language models (LLMs), enabling learning from outcome-level feedback across diverse tasks (Xu et al., 2025; WANG et al., 2025; Mroueh, 2025). In a nutshell, RLVR optimizes for end-task success, allowing models to explore novel reasoning strategies at scale (Zhao et al., 2025b; Ferrag et al., 2025; Su et al., 2025). Notably, it has been reported (Guo et al., 2025; Liu et al., 2025c; Zeng et al., 2025) that such training induces new emergent behaviors, such as self-reflection—the ability of a model to revisit, evaluate, and revise its prior outputs. For instance, the DeepSeek-R1 report (Guo et al., 2025) highlights that RLVR-trained models often generate tokens such as "wait", interpreted as signals of internal deliberation or critique.

Despite these observations, it remains unclear why such reflective behaviors emerge. Answering this question is crucial for understanding the foundations of reasoning in LLMs and guiding future methods for enhancing their performance. Moreover, empirical studies have shown that reflection correlates with more accurate and robust reasoning (Zuo et al., 2025; Yue et al., 2025; Liu et al., 2024a), and that prompting models to explicitly "wait" or reflect can further improve performance with test-time computing. However, this benefit may come at a cost: reflection can increase inference time, introduce unnecessary verbosity, and reduce computational efficiency (Yang et al., 2025b; Renze & Guven, 2024; Sui et al., 2025). These findings highlight not only the need to understand self-reflection in LLMs, but also the importance of controlling it to balance reasoning quality and efficiency. These gaps motivate this work to study the following research questions:

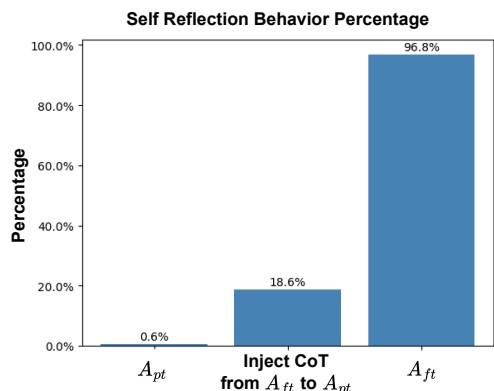

**Self Reflection Behavior Percentage**

100.0% — 96.8% (at $A_{ft}$)

18.6% (Inject CoT from $A_{ft}$ to $A_{pt}$)

0.6% (at $A_{pt}$)

**An Example of Self Reflection in $A_{pt}$**

**Question:** A strictly increasing sequence of positive integers $a_1, a_2, a_3, ...$ has the property that for every positive integer $k$, the subsequence $a_{2k-1}, a_{2k}, a_{2k+1}$ is geometric and the subsequence $a_{2k}, a_{2k+1}, a_{2k+2}$ is arithmetic. Suppose that $a_{13} = 2016$. Find $a_1$.
**Response:** Let's break down the problem step-by-step:
1. Understand the sequence properties:
… <omitted>…
8. **Reconsider** the relationship between geometric and arithmetic subsequences:
 … <omitted>…
14. Final Answer:After determining the correct $r$ and solving for $a_1$, we get:\boxed{2}

Figure 1: **Left:** Frequency distribution of self-reflection behaviors for pretrained model $\mathcal{A}_{\texttt{pt}}$, fine-tuned model $\mathcal{A}_{\texttt{ft}}$, and $\mathcal{A}_{\texttt{pt}}$ with reflection-inducing probing by injecting CoT from $\mathcal{A}_{\texttt{ft}}$, evaluated on the MATH500 dataset. **Right:** A representative example of spontaneous self-reflection in $\mathcal{A}_{\texttt{pt}}$, demonstrating that this capability emerges naturally during pretraining, albeit with different self-reflection tokens than those typically observed in RLVR fine-tuned models.

*Is self-reflection a novel behavior induced by RLVR, or does it already emerge during pretraining? Can we control self-reflection in LLMs to balance performance and computational efficiency?*

**Contribution**   In this work, we provide affirmative answers to both questions. First, we compare the reasoning behaviors of pretrained models and fine-tuned models (either via RLVR or distillation), and verify that self-reflection is already present in the pretrained model, albeit at a much lower frequency. Next, we analyze the hidden representations associated with reflective versus non-reflective reasoning, and find that they exhibit distinct activation patterns. Furthermore, we show that the degree of self-reflection can be modulated by a single direction in the representation space. Our contribution can be summarized as follows.

- **Self-Reflection already emerge during pretraining:** We demonstrate that self-reflection capabilities naturally exist in pretrained models and are not solely artifacts of RLVR. However, the frequency of such behavior is extremely low—for example, only 0.6% as shown in Figure 1. To isolate the model's capacity for self-reflection from its general reasoning ability, we propose a method, called *reflection-inducing probing*, that inserts reasoning traces—specifically, those that trigger self-reflection in a fine-tuned reasoning model—into the input of the pretrained model, and then measures whether the latter produces reflection in response. Using reflection-inducing probing, we observe that the pretrained model exhibits reflection with a frequency of 18.6%, significantly higher than the baseline of 0.6%, though still lower than the fine-tuned model (which is almost 100%). Through comparative analysis of hidden representations, we show that pre-trained models maintain internal structures that distinguish reflective behavior from non-reflective contexts—similar to fine-tuned models—further suggesting that pretrained models already possess self-reflection capabilities.

- **The degrees of self-reflection can be modulated by a single direction:** Motivated by the separability of reflective and non-reflective contexts in the hidden representation space, we use the method of difference-of-means (Rimsky et al., 2024) to construct a *self-reflection direction*, enabling control over self-reflection behavior for both pretrained and fine-tuned models. Our experiments demonstrate that this control mechanism offers a tunable trade-off between accuracy and efficiency: enhancing reflection improves accuracy by up to 12% on benchmarks, while suppressing it reduces output length by over 32% without significant performance degradation. We further show that this direction transfers robustly across diverse tasks—including mathematical and scientific reasoning—highlighting its universality as a shared, task-agnostic mechanism.

## 2 PRELIMINARY

### 2.1 TRANSFORMER LAYER

Decoder-only transformers (Liu et al., 2018) map input tokens $\mathbf{t} = (t_1, t_2, \ldots, t_n) \in \mathcal{V}^n$ to output probability distributions over the vocabulary $\mathcal{V}$. Let $\mathbf{h}_i^{(\ell)} \in \mathbb{R}^d$ denote the residual stream activation (also referred to as the hidden state) of the $i$-th token at the $\ell$-th layer, where $d$ is the dimensionality of the hidden state. Each of the $L$ transformer layers applies a sequence of attention and MLP transformations to update the residual stream:

$$\tilde{\mathbf{h}}_i^{(\ell)} = \mathbf{h}_i^{(\ell)} + \text{Attn}^{(\ell)}(\mathbf{h}_{1:i}^{(\ell)}), \quad \mathbf{h}_i^{(\ell+1)} = \tilde{\mathbf{h}}_i^{(\ell)} + \text{MLP}^{(\ell)}(\tilde{\mathbf{h}}_i^{(\ell)}) \tag{1}$$

The final hidden state is then projected to a probability distribution over the vocabulary $\mathcal{V}$ using an unembedding matrix followed by a softmax function.

### 2.2 SELF-REFLECTION

Recent work has shown that large language models (LLMs), even when pretrained purely on next-token prediction, demonstrate surprising levels of reasoning ability (Mondorf & Plank, 2024; Liu et al., 2024b; Wang et al., 2023). However, this capability can be significantly enhanced through fine-tuning on reasoning tasks using either reinforcement learning with verifiable rewards (RLVR) or supervised learning with distilled responses from reasoning models trained from RLVR (Liu et al., 2025a; Wang et al., 2024; Zhao et al., 2025a). We denote the pretrained model as $\mathcal{A}_{\text{pt}}$, and its fine-tuned variant as $\mathcal{A}_{\text{ft}}$.

A notable emergent behavior observed in fine-tuned models $\mathcal{A}_{\text{ft}}$ is self-reflection—the model's ability to internally evaluate, critique, or revise its own reasoning process. Unlike standard reasoning, which involves generating a direct solution to a task, self-reflection introduces an intermediate meta-cognitive step where the model pauses or backtracks to reconsider its prior outputs. This behavior is often marked by explicit tokens such as "wait," which have been shown to correlate with improved reasoning outcomes (Li et al., 2024; Liu et al., 2025c; Yeo et al., 2025).

Importantly, self-reflection is not limited to any specific model architecture, observed across both proprietary models (Jaech et al., 2024) and open-source systems (Guo et al., 2025; OLMo et al., 2024; Yang et al., 2025a), indicating that it may be a general emergent property of optimizing for complex reasoning objectives. While models may signal self-reflection using various phrases, including "wait", "let me double-check", or "I might have made a mistake", we focus our analysis on the canonical token "wait" due to its high frequency and clear association with reflective behavior. Our analysis confirms that "wait" is the most commonly used reflection marker across the DeepSeek-R1 series of models (Guo et al., 2025). Supporting analyses can be found in Appendix D. Crucially, the analytical framework we develop can generalize beyond "wait", extending to any token that plays an analogous reflective role within the reasoning trajectory.

## 3 SELF-REFLECTION ALREADY EMERGES DURING PRETRAINING

In this section, we conduct a systematic analysis of self-reflection behavior in both pretrained models $\mathcal{A}_{\text{ft}}$ and fine-tuned ones $\mathcal{A}_{\text{pt}}$. Using the MATH500 dataset (Hendrycks et al., 2021) as our evaluation benchmark, we compare DeepSeek-R1-Distill-Qwen-1.5B ($\mathcal{A}_{\text{ft}}$), a model fine-tuned from Qwen2.5-1.5B ($\mathcal{A}_{\text{pt}}$) (Guo et al., 2025; Yang et al., 2024). First, we show that self-reflection naturally emerges in pretrained models, though at a substantially lower frequency than in RLVR-distilled counterparts. Second, through analysis of hidden state representations, we find that even pretrained models implicitly encode and differentiate between self-reflective and nonself-reflective states, despite rarely generating reflective outputs explicitly.

### 3.1 PROBING SELF-REFLECTION IN PRETRAINED MODELS

To investigate whether self-reflection emerges intrinsically in $\mathcal{A}_{\text{pt}}$, rather than being solely a byproduct of fine-tuning strategies such as RLVR, we examined the behavior of $\mathcal{A}_{\text{pt}}$ on mathematical reasoning tasks using the MATH500 benchmark. Figure 1 (left) highlights the contrast in self-reflection

**Question**

Convert the point $(0, 3)$ in rectangular coordinates to polar coordinates. Enter your answer in the form $(r, \theta)$, where $r > 0$ and $0 \leq \theta < 2\pi$.

**Response**

Okay, so I need to convert the <omitted> So, plugging in the values, tan $\mu$ = 3/0.**\n\n Wait,** division by zero? That's undefined. <omitted> **But wait,** let me think again.**\n\n**Since the point is (0, 3), <omitted> **but** since the point is in the positive y-axis, we need to choose $\theta$ = π/2. <omitted> Therefore, the polar coordinates are (3, π / 2).**\n\n Wait** a minute, let me visualize this to ensure I'm correct <omitted> I remember that occurs at $\theta$ = π/2 and $\theta$ = 3π/2. **But wait,** just to make sure, let me think about another approach <omitted> Therefore, the polar coordinates are \boxed{3, π/2}

R — reflection inducing token

N — non-reflection inducing token

W — self-reflection token

Figure 2: **Hidden State Selection Methodology:** We identify tokens immediately preceding "wait" tokens as reflection-inducing tokens and extract their hidden states. For comparison, we collect hidden states of identical tokens appearing in non-reflective contexts. This contrastive approach enables us to analyze the neural signatures associated with self-reflection in language models.

frequency between $\mathcal{A}_{\mathtt{pt}}$ and $\mathcal{A}_{\mathtt{ft}}$ models, while the right panel shows a representative instance of naturally occurring self-reflection in $\mathcal{A}_{\mathtt{pt}}$. Remarkably, even in the absence of task-specific supervision, $\mathcal{A}_{\mathtt{pt}}$ exhibited spontaneous self-reflective behavior in a small but non-negligible fraction of cases—approximately 0.6%, as shown in Figure 1. These instances are characterized by explicit reconsideration or revision of prior reasoning steps. For details on how we identify self-reflection instances, please see Appendix D. While the self-reflection tokens differ somewhat from those typically observed in RLVR-trained models, their reflective nature is still discernible. These findings suggest that self-reflection is not solely acquired through fine-tuning, but rather emerges as a latent capability within the $\mathcal{A}_{\mathtt{pt}}$—one that is infrequently activated but nonetheless present.

**Reflection-Inducing Probing by Injecting CoT from $\mathcal{A}_{\mathtt{ft}}$ into $\mathcal{A}_{\mathtt{pt}}$** However, the extremely low frequency of self-reflection in $A_{\mathtt{pt}}$ makes it challenging to analyze systematically and to develop methods (which will be studied in the next section) for controlling such behavior. To address this challenge, we propose a probing method, termed *reflection-inducing probing*, that isolates the model's capacity for self-reflection from its general reasoning ability. The key idea is to decouple reasoning competence from reflective behavior by inserting reasoning traces generated by the fine-tuned model $\mathcal{A}_{\mathtt{ft}}$ into the input of the pretrained model $\mathcal{A}_{\mathtt{pt}}$, and then measuring whether $\mathcal{A}_{\mathtt{pt}}$ generates reflection in response.

Formally, given a question $q$, we use the fine-tuned model $\mathcal{A}_{\mathtt{ft}}$ to generate a sequence of reasoning tokens:

$$\mathcal{A}_{\mathtt{ft}}(q) = (\ \underbrace{r_1}_{\text{pre-reflection}}\ ,\ \underbrace{\mathtt{reflection}}_{\text{explicit signal}},\ \underbrace{r_2}_{\text{post-reflection}}\ )$$

where $r_1$ denotes the initial chain-of-thought leading up to an explicit reflection token (e.g., "wait"), and $r_2$ represents the revised or continued reasoning after reflection. We then construct a new prompt by inserting $r_1$ (the pre-reflection reasoning) into the input of $\mathcal{A}_{\mathtt{pt}}$, and evaluate whether $\mathcal{A}_{\mathtt{pt}}$ independently produces a reflection token at the appropriate point. This setup ensures that both models operate on similar reasoning contexts, eliminating confounding differences in reasoning capability. By comparing the frequency and consistency of self-reflection under this controlled setting, we can more directly assess whether reflective behavior is present in the pretrained model and to what extent it is amplified by fine-tuning. The frequency of generating reflection with reflection-inducing probing is reported in Figure 1.

**Self-Reflection emerges naturally in pretrained models albeit with much lower frequency** Remarkably, $\mathcal{A}_{\mathtt{pt}}$ exhibits clear self-reflective behavior in 18.6% of these cases—a dramatic increase from its baseline rate. This differential response demonstrates that while $\mathcal{A}_{\mathtt{pt}}$ rarely produces overt reflection markers in standard contexts, it possesses latent self-reflection capabilities that can be activated by appropriate contextual triggers. These findings strongly suggest that self-reflection mechanisms are encoded during pretraining, rather than being exclusively developed

through reinforcement learning. With 18.6% self-reflection cases, the subsequent section analyzes the hidden state representations underlying these self-reflective behaviors to provide further insights.

## 3.2 Hidden State Representations of Self-Reflection

To further investigate the emergence of self-reflection, we analyze the internal representations of the model when it decides to generate reflection versus when it does not. Specifically, we focus on the hidden states associated with reasoning tokens that immediately precede the generation of a reflection token (e.g., "wait"), and compare them to those that do not lead to reflection.

Since both the pretrained model $\mathcal{A}_{\texttt{pt}}$ and the fine-tuned one $\mathcal{A}_{\texttt{ft}}$ exhibit self-reflection behaviors, we use $\mathcal{A}$ to denote a generic model (either pretrained or fine-tuned), which will be specified in context. Given a question $q$, suppose the model $\mathcal{A}$ generates a sequence of reasoning tokens that includes self-reflection. Let $r = \mathcal{A}(q) = (r_1, \texttt{reflection}, r_2)$, where $r_1$ precedes the reflection token and $r_2$ follows it. Due to the auto-regressive nature of transformer models, the information from the question $q$ and the reasoning tokens $r_1$ is aggregated into the hidden representation of the final token in $r_1$, which is then used by the last layer to predict the next token—the reflection token. For convenience, we refer to the final token in $r_1$ as a *reflection-inducing token*, though its hidden state captures information from the entire preceding context $(q, r_1)$. Reflection-inducing tokens often coincide with sentence-final punctuation (e.g., ".", "!", or closing brackets) or specific markers such as "But". Now with a slight abuse of notation, let $h^{(\ell)}_{\text{reflection-inducing}}(q, r)$ denote the $\ell$-th layer hidden state of a model at $\mathcal{A}$ corresponding to the reflection-inducing token. We collect all such hidden states from model outputs that contain reflection tokens into the following set

$$\mathbb{H}^{(\ell)}_{\texttt{reflect}} = \left\{ h^{(\ell)}_{\text{reflection-inducing}}(q, r) \in \mathbb{R}^d \right\}. \tag{2}$$

To study the properties of hidden states associated with self-reflection, we contrast this set with representations from cases where the model *does not* generate reflection. Specifically, to eliminate the confounding effect of token surface form, we extract hidden states from tokens that share the same form as reflection-inducing tokens (e.g., sentence-final punctuation), but which do not lead to self-reflection in the subsequent responses (within 100 tokens in the experiments). For notational convenience, we refer to these as *non-reflection-inducing tokens*, and denote their corresponding hidden states as $h^{(\ell)}_{\text{non-reflection-inducing}}(q, r)$. See Figure 2 for an illustration comparing reflection-inducing and non-reflection-inducing tokens. We collect such non-reflection-inducing into the following set

$$\mathbb{H}^{(\ell)}_{\texttt{non-reflect}} = \left\{ h^{(\ell)}_{\text{non-reflection-inducing}}(q, r) \in \mathbb{R}^d \right\}. \tag{3}$$

This design ensures a fair comparison by controlling for the surface form of the reflection-inducing token, ensuring that any differences in hidden representations are attributable to the model's decision to reflect.

For $\mathcal{A}_{\texttt{pt}}$ models, which rarely generate self-reflective outputs, we use the method of reflection-inducing probing by injecting CoT from $\mathcal{A}_{\texttt{pt}}$ into $\mathcal{A}_{\texttt{ft}}$ to elicit reflective behavior. To visualize these high-dimensional representations, we employ principal component analysis (PCA) for dimensionality reduction, projecting the hidden states into a 2D space. Figure 3 presents the visualizations for the 15th layer (out of 28 total layers) for both models, with more layers presented in Appendix C.

Our analysis reveals a pattern: **both models show clear separation between self-reflection and nonself-reflection states**. While this separation is expected in $\mathcal{A}_{\texttt{ft}}$, which was explicitly trained to exhibit self-reflective behavior, the equally distinct clustering in $\mathcal{A}_{\texttt{pt}}$ is remarkable. Despite rarely generating explicit self-reflection tokens in its outputs, $\mathcal{A}_{\texttt{pt}}$ maintains internal representations that clearly distinguish between self-reflective and nonself-reflective contexts. This finding provides strong evidence that self-reflection capabilities develop during pretraining, with models encoding these patterns in their hidden state representations even when they rarely manifest in generated text. We will exploit this internal structure to develop methods for controlling self-reflection in the next section.

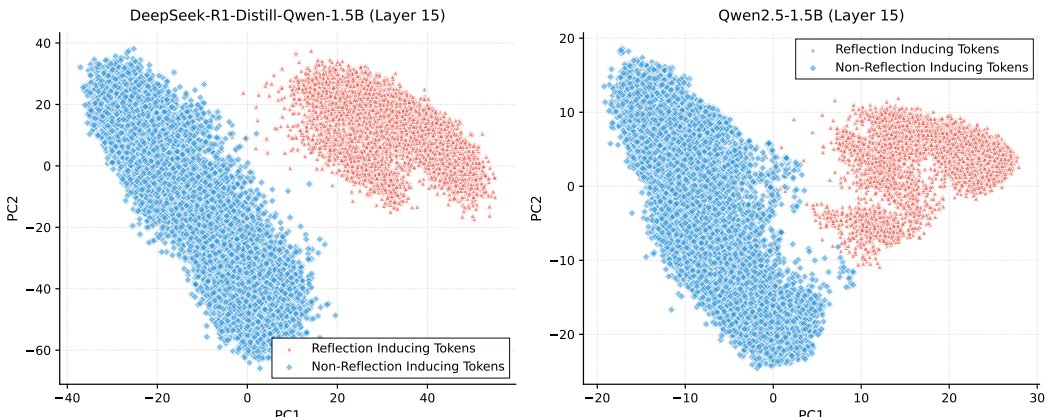

Figure 3: **PCA visualization of hidden state representations of layer-15 for DeepSeek-R1-Distill-Qwen-1.5B and Qwen2.5-1.5B. Both models show separation between $\mathbb{H}^{(\ell)}_{\texttt{reflect}}$ and $\mathbb{H}^{(\ell)}_{\texttt{non-reflect}}$.**

## 4 CONTROLLING SELF-REFLECTION IN LANGUAGE MODELS

In this section, we introduce our approach for identifying and manipulating self-reflection vectors in LLMs. Building on our finding in the last section that hidden representations distinctly separate self-reflective from non-reflective contexts, we construct self-reflection vectors, directions in activation space associated with self-reflective reasoning. We then demonstrate how these vectors can be used to bidirectionally control self-reflection behavior, either enhancing it to improve reasoning accuracy or suppressing it to reduce computational overhead. Through extensive evaluation across multiple mathematical reasoning benchmarks, we show that our method significantly outperforms strong baselines while offering flexible control over the performance-efficiency trade-off. Finally, we examine the cross-domain transferability of these vectors, revealing their potential as universal controls for self-reflection across diverse reasoning tasks.

### 4.1 EXTRACT SELF-REFLECTION VECTORS

To identify the self-reflection vector in the residual stream activations, we compute the difference between the activations of self-reflective and nonself-reflective contexts. This technique, known as difference-in-means, effectively isolates key feature directions, as demonstrated in prior work (Rimsky et al., 2024; Arditi et al., 2024; Wu et al., 2025), motivating our application of this approach to the self-reflection domain. As in Section 3, we focus on the hidden state of reflection-inducing token, positing that this state encodes the model's transition into self-reflection reasoning.

For each layer $\ell \in \{1, \ldots, L\}$, we compute the mean hidden states in the reflection set $\mathbb{H}^{(\ell)}_{\texttt{reflect}}$ and non-reflection set $\mathbb{H}^{(\ell)}_{\texttt{non-reflect}}$ as

$$\boldsymbol{\mu}^{(\ell)}_{\texttt{reflect}} = \text{mean}(\mathbb{H}^{(\ell)}_{\texttt{reflect}}), \quad \boldsymbol{\mu}^{(\ell)}_{\texttt{non-reflect}} = \text{mean}(\mathbb{H}^{(\ell)}_{\texttt{non-reflect}}), \tag{4}$$

and then construct the self-reflection vector as the difference-in-means vector

$$\boldsymbol{v}^{(\ell)} = \boldsymbol{\mu}^{(\ell)}_{\texttt{reflect}} - \boldsymbol{\mu}^{(\ell)}_{\texttt{non-reflect}}, \tag{5}$$

which captures both the direction along which self-reflective and nonself-reflective activations diverge, and the magnitude of that divergence.

### 4.2 MODEL INTERVENTIONS FOR CONTROLLING TRADE-OFF BETWEEN REASONING AND EFFICIENCY

To actively modulate a model's tendency to reflect, motivated by the linear representation hypothesis and prior work (Arditi et al., 2024), we apply simple linear interventions based on the self-reflection vector $\boldsymbol{v}^{(\ell)}$ extracted from the $\ell$-th layer, which is expected to capture the direction in representation

space most associated with self-reflection. Specifically, we modify each residual stream $\boldsymbol{h}^{(\ell)}$ at the $\ell$-th layer in (1) according to

$$\widehat{\boldsymbol{h}}^{(\ell)} = \boldsymbol{h}^{(\ell)} + \alpha \boldsymbol{v}^{(\ell)} \left\langle \boldsymbol{h}^{(\ell)}, \boldsymbol{v}^{(\ell)} \right\rangle, \tag{6}$$

where $\widehat{\boldsymbol{h}}^{(\ell)}$ then replaces $\boldsymbol{h}^{(\ell)}$ as the input to the next layer, and the scalar $\alpha$ controls the strength of the intervention. When $\alpha > 0$, the model's self-reflection behavior is enhanced; when $\alpha < 0$, it is suppressed. Setting $\alpha = 0$ disables the intervention, preserving the model's default behavior.

**Ablation study on $\alpha$**   To illustrate the effect of the linear intervention method for controlling self-reflection, we conduct an ablation study by varying the self-reflection steering strength $\alpha$ from $-1.0$ to $1.0$, injecting the reflection vector at layer 14 in DeepSeek-R1-1.5B on the MATH-500 benchmark. The result is shown in Figure 12. Negative $\alpha$ values shorten responses, reducing average token length, while preserving accuracy. In contrast, positive $\alpha$ both lengthens responses, and boosts performance, peaking at $\alpha$=0.03 with a 12% performance gain in Pass@1, before declining at larger values due to over-reflection. This clear trade-off underscores $\alpha$ as a practical knob for balancing verbosity against reasoning depth. Further ablation detail on the effect of the injection layer is provided in Appendix E.

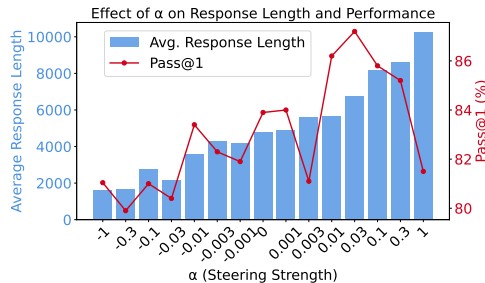

Figure 4: Effect of $\alpha$ on performance and response length on Math500 Dataset (reflection vector is injected on layer 14).

### 4.3 EXPERIMENTAL RESULTS

We evaluate our self-reflection control mechanism on two mathematical reasoning benchmarks, MATH-500 (Hendrycks et al., 2021) and AIME 2024, and one scientific QA benchmark, GPQA Diamond (Rein et al., 2024). Experiments are conducted using DeepSeek-R1 and Qwen2.5 models at both 1.5B and 7B parameter scales (Guo et al., 2025; Yang et al., 2024). To demonstrate that our method is not limited to these architectures, we also evaluated it on OLMO-2-13B-Instruct and Llama 3.1 8B Instruct (Grattafiori et al., 2024; OLMo et al., 2024).

We compare three inference strategies: **Vanilla (Baseline)**, which uses standard setting without any intervention; **BF (Budget Forcing)**, which enforces reflection by appending a "wait" token at the end of initial short generations (Guo et al., 2025; Muennighoff et al., 2025); and **Self-Reflection (SR) Enhanced/Suppressed**, our proposed technique that perturbs hidden states using self-reflection vectors scaled by a coefficient $\alpha$ (positive for enhancement, negative for suppression). Notably, $\mathcal{A}_{\mathtt{ft}}$ can frequently trigger self-reflection, so we apply both SR enhancement and suppression; whereas $\mathcal{A}_{\mathtt{pt}}$ trigger it only rarely, and we thus evaluate SR enhancement only. For details on selecting the optimal injection strategy, please refer to Appendix E.

**Key Findings.**   Our results highlight three major insights. First, SR Enhancement improves reasoning performance across most evaluated datasets and model sizes. For example, Qwen2.5 7B's performance on MATH-500 jumps by 12.0 percentage points (from 44.6% to 56.6%), and DeepSeek-R1 variants enjoy similar boosts when employing reflection-enhanced decoding. Furthermore, this performance gain is also accompanied by a noticeable increase in response length, suggesting that longer responses can be beneficial when tackling more challenging reasoning tasks.

Second, SR Suppression offers fine-grained control over computational cost. It consistently reduces output length—often by more than 50%—while preserving most of the model's accuracy. Notably, DeepSeek-R1-7B reduces average token length from 3564 to 2451 on MATH-500 with only a minor drop in Pass@1, which remains above 91%.

Finally, these effects demonstrate strong generalizability across training paradigms and model families. The observed improvements hold for both $\mathcal{A}_{\mathtt{ft}}$ models (e.g., DeepSeek-R1, OLMO2) and $\mathcal{A}_{\mathtt{pt}}$ models

| Size | Method | MATH-500 | | AIME 2024 | | GPQA Diamond | |
|---|---|---|---|---|---|---|---|
| | | Pass@1 ↑ | LEN ↓ | Pass@1 ↑ | LEN ↓ | Pass@1 ↑ | LEN ↓ |
| DeepSeek-R1-1.5B | Vanilla | 84.1 | 4755 | 29.2 | 6118 | 14.0 | 4250 |
| | BF | 85.5 | 10122 | 30.0 | 8986 | 14.8 | 5210 |
| | SR Enhanced | **87.4** | 9458 | **33.5** | 8132 | **18.9** | 7496 |
| | SR Suppressed | 83.2 | **3716** | 27.3 | **5229** | 13.8 | **3795** |
| DeepSeek-R1-7B | Vanilla | 92.7 | 3585 | 55.8 | 4558 | 27.1 | 3696 |
| | BF | 93.1 | 7111 | 54.5 | 9629 | 32.0 | 5802 |
| | SR Enhanced | **93.5** | 8959 | **58.2** | 5684 | **34.6** | 6513 |
| | SR Suppressed | 91.2 | **2439** | 52.9 | **3319** | 26.5 | **3120** |
| OLMo2-13B | Vanilla | 41.8 | 3075 | 7.2 | 3753 | 15.2 | 2645 |
| | BF | 43.2 | 4891 | 10.2 | 4894 | 17.0 | 3714 |
| | SR Enhanced | **47.9** | 4574 | **11.7** | 4941 | **19.1** | 3490 |
| | SR Suppressed | 40.8 | **2106** | 6.5 | **2656** | 13.7 | **1375** |
| Qwen2.5 1.5B | Vanilla | 27.5 | **1515** | 0.2 | **544** | 5.7 | **702** |
| | BF | 29.6 | 3993 | 0.1 | 4517 | 5.0 | 1389 |
| | SR Enhanced | **36.9** | 1836 | **6.1** | 2525 | **6.8** | 814 |
| Qwen2.5 7B | Vanilla | 44.8 | **1294** | 3.9 | **2285** | 16.2 | **598** |
| | BF | 46.0 | 3986 | 5.3 | 4526 | **16.9** | 1639 |
| | SR Enhanced | **56.8** | 2671 | **16.1** | 2941 | 14.8 | 1816 |
| Llama 3.1 8B | Vanilla | 44.0 | **1613** | 3.5 | **1653** | 30.9 | **1914** |
| | BF | 40.1 | 3771 | 2.9 | 3912 | 26.1 | 5031 |
| | SR Enhanced | **57.7** | 2887 | **16.5** | 2974 | **34.1** | 3995 |

Table 1: Performance across mathematical and scientific reasoning benchmarks using models of different sizes. We compare three inference strategies: **Vanilla** (no intervention), **BF** (budget forcing via "wait" token insertion), and our method: **SR Enhanced/Suppressed** (applying positive or negative $\alpha$ to respectively amplify or suppress self-reflection during inference). Pass@1 indicates accuracy (higher is better); LEN indicates average generation length (lower is better). We use 10% of the data as a validation set to select the optimal $\alpha$ in Eq. 6.

(e.g., Qwen2.5, Llama3.1), suggesting that the self-reflection signal is a robust and transferable mechanism that transcends specific architectures or fine-tuning methods.

Building on this generalizability, we emphasize the practical value of inference-time control over latent self-reflection dynamics. Unlike rigid interventions such as budget forcing, our method affords semantically grounded, continuous modulation of a model's internal self-reflection, enabling a tunable trade-off between performance and efficiency, especially in resource-constrained settings.

**To verify that the observed gains are not due to chance, we perform a paired significance analysis over individual problems for every model–dataset combination. Table 2 reports one-sided McNemar-style $p$-values comparing SR-Enhanced to Vanilla. In 15 out of 18 cases, the $p$-value is below 0.05, closely mirroring the accuracy differences in Table 1 and indicating that the improvements are statistically reliable. The remaining three cases, DeepSeek-R1-7B on MATH-500 and two GPQA settings for the smaller Qwen models, are exactly those where SR-Enhanced offers little or no advantage, reinforcing that we do not overstate effects in regimes with negligible gains.**

| Dataset | DeepSeek-R1-1.5B | DeepSeek-R1-7B | OLMo2-13B | Qwen2.5-1.5B | Qwen2.5-7B | Llama 3.1-8B |
|---|---|---|---|---|---|---|
| MATH-500 | 0.0057 | 0.2125 | $4.0 \times 10^{-4}$ | $2.0 \times 10^{-4}$ | $4.0 \times 10^{-5}$ | 0.0017 |
| AIME 2024 | 0.0066 | 0.0139 | 0.0057 | $1.0 \times 10^{-4}$ | $2.0 \times 10^{-5}$ | $7.0 \times 10^{-4}$ |
| GPQA Diamond | $2.0 \times 10^{-4}$ | $5.0 \times 10^{-4}$ | 0.0030 | 0.0519 | 0.8665 | 0.0334 |

Table 2: **One-sided $p$-values from a McNemar-style paired test over problems, comparing SR-Enhanced vs. Vanilla for each model and dataset.**

## 4.4 TRANSFERABILITY OF SELF-REFLECTION VECTORS

To investigate the transferability of self-reflection vectors across different reasoning domains, we evaluated whether vectors extracted from the GPQA Diamond dataset could be effectively trans-

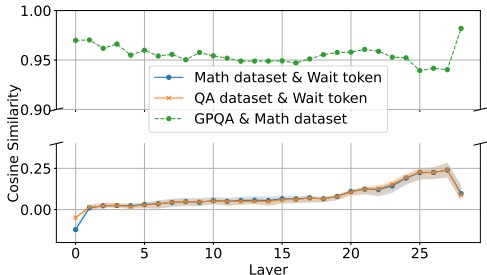

| Model | Setting | PASS@1 | Len |
|---|---|---|---|
| 1.5B | SR Enhanced | 86.4 | 7781 |
| | SR Suppressed | 82.6 | 3684 |
| 7B | SR Enhanced | 93.0 | 6007 |
| | SR Suppressed | 91.0 | 2992 |

Figure 5: **Left:** Cosine similarity of self-reflection vectors and the "wait" token across MATH500 and GPQA datasets. The green curve shows similarity between vectors from MATH500 and GPQA. Blue and orange curves show similarity with the "wait" token. **Right:** Performance on MATH500 when applying self-reflection vectors extracted from GPQA Diamond to DeepSeek-R1 models.

ferred to mathematical reasoning tasks in MATH500. We compute the cosine similarity between self-reflection vectors extracted from different domains (MATH500 and GPQA Diamond), and between these vectors and the embedding of the token "wait" in DeepSeek-R1-Distill-Qwen-1.5B. For tokenizers containing multiple subword tokens for "wait", we report the average cosine similarity along with its variance. The results are plotted in Figure 5(left). Our analysis revealed remarkable consistency in the neural signatures of self-reflection across these distinct domains. Specifically, vectors extracted from GPQA and MATH500 exhibit high cosine similarity, suggesting that the internal representation of reflective states is largely domain-invariant. Notably, these self-reflection vectors are substantially different from the embedding of the token "wait", indicating that they encode deeper semantic properties of reflective behavior rather than surface-level cues.

Further, we evaluate the performance of self-reflection vectors derived from GPQA-Diamond on MATH500 using our proposed intervention method, SR-Enhanced/Suppressed, and present the results in Figure 5(right). Notably, we observe similar performance gains to those seen with in-domain self-reflection vectors, as reported in Table 1. This cross-domain transfer demonstrates that the reflective mechanism captures a generalizable cognitive pattern rather than being confined to task-specific reasoning strategies. Together, these findings suggest that LLMs develop a unified internal representation of self-reflection, one that can be leveraged across tasks without the need for domain-specific fine-tuning.

## 4.5 COMPARISON OF DIFFERENT METHODS

**To validate our choice of the difference-in-means approach for generating steering vectors, we conduct a comparative analysis against alternative feature-extraction techniques. We consider four variants: (i) a difference-in-means direction with the projected update used in the main paper, (ii) a difference-in-means direction with additive steering, (iii) a standard PCA direction, and (iv) contrastive PCA (Abid et al., 2018). All methods are evaluated under identical intervention settings for both enhancing and suppressing self-reflection.**

Table 3: Comparison of feature-extraction methods and steering methods for constructing the self-reflection vector on DeepSeek-R1 1.5B.

| Method | SR Enhanced | | SR Suppressed | |
|---|---|---|---|---|
| | PASS@1 ↑ | LEN ↓ | PASS@1 ↑ | LEN ↓ |
| Difference-in-means | **87.2** | **9420** | **83.4** | **3738** |
| ↪ **Additive Steering** | 86.0 | 9845 | 83.0 | 3639 |
| PCA | 84.3 | 10074 | 79.1 | 3912 |
| Contrastive PCA | 84.7 | 9876 | 80.6 | 4078 |

**The evaluation was performed under identical intervention settings for both the enhancement and suppression of the target behavior. The quantitative results are shown in Table 3. The data indicate that the difference-in-means approach achieves the highest task performance in both intervention scenarios, yielding superior PASS@1 scores while maintaining reasonable vector lengths.**

**This empirical finding aligns with insights from recent work in mechanistic interpretability (Wu et al., 2025). These studies suggest that for the specific goal of causal steering of model behavior, simple directional vectors derived from differences often outperform other techniques, thereby motivating our choice of the difference-in-means approach.**

## 4.6 ADDITIONAL ANALYSIS OF SELF-REFLECTION TOKENS ON LLAMA 3.1 8B

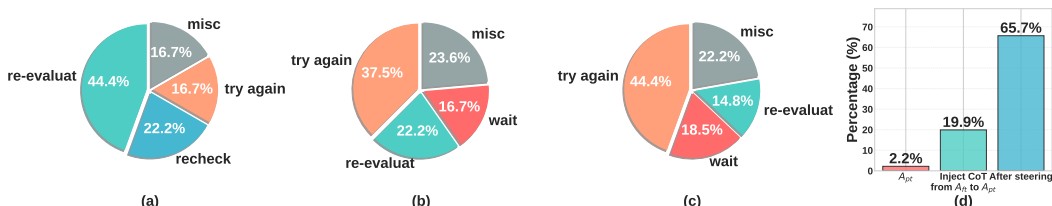

Figure 6: Additional analysis of self-reflection tokens on Llama 3.1 8B. (a)–(b): distributions of frequent reflection markers before and after injecting CoT from $\mathcal{A}_{\tt ft}$ into $\mathcal{A}_{\tt pt}$. (c): distributions of frequent reflection markers after steering in $\mathcal{A}_{\tt ft}$ with the self-reflection vector. (d): percentage that contains at least one reflection token, comparing $\mathcal{A}_{\tt pt}$, after injecting CoT from $\mathcal{A}_{\tt ft}$ into $\mathcal{A}_{\tt pt}$ and steered $\mathcal{A}_{\tt ft}$.

In addition to the analysis on Qwen models, we investigate whether the learned self-reflection direction captures a model-agnostic behavioral pattern. To this end, we replicate our token-level study on Llama 3.1–8B and measure how steering $\mathcal{A}_{\tt pt}$ using CoT traces extracted from $\mathcal{A}_{\tt ft}$ reshapes the distribution of explicit reflection markers.

Figure 6 provides four complementary views of the effect. Panels (a) and (b) show the empirical frequency of the most common reflection-like tokens before and after intervention. Panel (c) visualizes the token distribution induced directly by the steering vector itself after injection. Panel (d) further summarizes the fraction of generated answers that contain at least one reflection token.

Across all four panels, we observe a consistent trend: steering substantially increases the usage of a diverse set of reflection markers, including wait, re-check, re-evaluate, and try again, yet no single token dominates. This distributed increase indicates that the direction is not memorizing or over-amplifying any specific token. Instead, the intervention activates a broader functional mode associated with reflective reasoning.

The behavior in Panel (c) is particularly revealing. Even though the steering vector is constructed solely from hidden states associated with the token wait, the resulting intervention increases the usage of multiple distinct reflection markers rather than merely reproducing wait. This demonstrates that the learned direction generalizes beyond surface-level token identity: it stimulates a latent mechanism for self-checking and revision, rather than a token-specific association.

Taken together with our Qwen results, the cross-model consistency and cross-token generality provide strong evidence that the learned direction captures an underlying functional mode of self-reflection. Rather than encoding a single marker, it induces a model-independent shift toward reflective reasoning behavior.

## 5 CONCLUSION

In this paper, we demonstrated that self-reflection in large language models is an emergent capability that develops during pretraining rather than being uniquely induced by reinforcement learning techniques. Through contrastive analysis of hidden state representations, we revealed that even models with minimal explicit reflection behavior maintain internal neural signatures that distinguish self-reflective contexts. By exploiting these representations, we developed an intervention method that enables bidirectional control over self-reflection, providing a flexible mechanism to navigate the performance-efficiency trade-off without requiring additional training.

## REPRODUCIBILITY STATEMENT

We are committed to ensuring the reproducibility of our results. All implementation details, including model configurations, and evaluation procedures, are provided in Appendix A. We will also release our code and scripts upon publication to facilitate replication and further research.

## ETHICS STATEMENT

This work investigates the mechanisms of self-reflection in LLMs and introduces methods to amplify or suppress reflective reasoning behaviors. While such techniques may contribute to a deeper understanding of model internals and enable improvements in reasoning quality, they could also be misapplied to manipulate model behaviors in unintended or undesirable ways. We therefore emphasize that our contributions are intended for research purposes, and we encourage responsible use of these findings in line with ethical standards for AI development.

## USE OF LLMS

We used LLMs to assist in the preparation of this paper, primarily for polishing writing, improving readability, and clarifying technical descriptions. All research questions, methods, and analyses were designed and conducted by the authors.

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

## A    IMPLEMENTATION DETAILS

For the DeepSeek-R1-Distill-Qwen models, we adopted a specialized prompting strategy that incorporates the explicit token `<think>` to elicit self-reflective reasoning and promote internal deliberation.

In contrast, for the Qwen models, we employed the same prompt template but omitted the `<think>` token. This design allowed us to isolate and assess the specific influence of `<think>` on eliciting reflective behaviors and its downstream impact, as Qwen models do not natively rely on such explicit triggers.

To support complex multi-step reasoning, we set the maximum generation length to 32,784 tokens across all experiments, ensuring that outputs were not prematurely truncated. All experiments were conducted on a computing cluster equipped with 8 NVIDIA A5000 GPUs.

## B    RELATED WORK

### B.1    FEATURES AS DIRECTIONS

Extracting feature directions, often derived from contrastive pairs of inputs, is an established technique for analyzing and manipulating neural network representations (Rimsky et al., 2024; Burns et al., 2022; Zou et al., 2023). It is widely recognized that adding such feature vectors to the model's residual stream can modify its behavior, although the optimal intervention points and specific methodologies remain areas of active research (Von Rütte et al., 2024; Jorgensen et al., 2023).

Several studies suggest that directions within the activation space capture semantic features more effectively or interpretably than individual neurons (Geiger et al., 2024; Park et al., 2023; Bolukbasi et al., 2016). Recent approaches utilize techniques like sparse autoencoders to discover these feature directions in an supervised manner (Huben et al., 2024). However, alternative methods such as Difference-in-Means (DiffMean) (Arditi et al., 2024) have demonstrated strong performance, sometimes exceeding that of sparse autoencoders, in specific applications like concept detection and model steering (Wu et al., 2025). Furthermore, the underlying assumption that features can be represented linearly has proven effective in tasks such as targeted concept erasure within language models (Shao et al., 2023; Belrose et al., 2023; Feng et al., 2025).

### B.2    SELF-REFLECTION IN LANGUAGE MODELS

The concept of self-reflection in language models has gained increasing attention as a mechanism for improving reasoning quality and alignment. Recent studies (Lightman et al., 2024; Madaan et al., 2023; Puerto et al., 2024; Zelikman et al., 2022; Lightman et al., 2023; Li et al., 2025) have explored how prompting models to generate intermediate reflections, critiques, or alternative solutions can improve final outputs in tasks such as math problem solving, programming, and factual reasoning. While many of these techniques are implemented at the prompting level or through chain-of-thought scaffolding, they suggest that self-reflection is a powerful tool for enhancing reasoning. Notably, these methods often induce substantial increases in generation length and latency, raising questions about the trade-off between deliberation and efficiency (Yang et al., 2025b; Yi & Wang, 2025; Chen et al., 2024; Team et al., 2025).

Recent studies have show that RLVR (Shah et al., 2025; Shinn et al., 2023; Wang et al., 2025; Xu et al., 2025) can improve reasoning abilities by explicitly training models to reflect using outcome-based feedback. Empirically, trained models (Guo et al., 2025; Liu et al., 2025b) like DeepSeek-R1 demonstrate significant improvements over baseline models in mathematical and logical reasoning tasks, and showcase new emergent behaviors such as self-reflection. Our work shows that self-reflection is a broadly distributed and latent feature of LLMs, not exclusively a product of RLVR. Concurrent works (Shah et al., 2025; Yue et al., 2025) also suggest that RLVR does not necessarily introduce novel reasoning abilities beyond those acquired during pretraining; instead, it primarily serves to amplify abilities already present in the model. Our work also complements this literature by showing that LLMs already encode latent self-reflection signals in their hidden states—even in models not explicitly trained for such behavior—and that reflection can be selectively enhanced or suppressed through lightweight vector interventions. This enables fine-grained control over reflective

behavior, including the ability to mitigate over-reflection, thereby avoiding unnecessary computational overhead without sacrificing performance.

## C   PCA VISUALIZATION OF SELF-REFLECTION STATES

**To investigate how self-reflective states are internally represented, we apply PCA to hidden states extracted from Qwen2.5-1.5B and DeepSeek-R1-Distill-Qwen-1.5B. Figure 3 in the main text shows the PCA projection of the final-layer representations, while Figures 8 and 9 display representative layers for Qwen2.5-1.5B and DeepSeek-R1-Distill-Qwen-1.5B, respectively.**

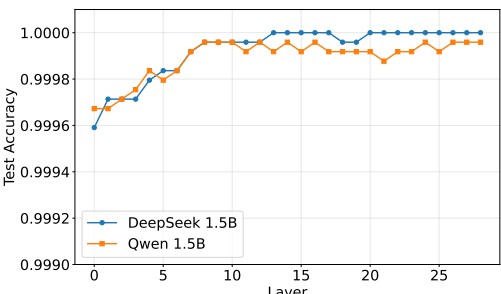

Across layers, reflection-inducing and non–reflection-inducing tokens form two elongated clouds that are increasingly separated along the first principal component. In very early layers there is some overlap, but from middle layers onward a clear margin emerges, and by the deepest layers the two populations are well separated for both model families. This progression indicates that the model progressively sharpens a reflection-related direction in representation space rather than encoding it only at a single depth.

Figure 7: **Linear-probe test accuracy of a linear SVM trained to distinguish reflection-inducing tokens from non–reflection-inducing tokens at each layer of DeepSeek-R1-Distill-Qwen-1.5B and Qwen2.5-1.5B. Accuracy is above 99% across all layers for both models, showing that the two groups are almost perfectly linearly separable.**

**The PCA visualizations are consistent with our quantitative linear-probe results in Figure 7, where a linear SVM trained on 10% of the hidden states and evaluated on the remaining 90% achieves above 99% test accuracy at every layer for both models, confirming that reflection-inducing and non–reflection-inducing tokens are almost perfectly linearly separable in the original high-dimensional space. Notably, this linear structure also appears in Qwen2.5-1.5B, which emits very few explicit self-reflection tokens, supporting our claim that self-reflection is encoded as a latent representational direction rather than being tied only to surface markers or a specific training recipe.**

**These geometric patterns help explain why our steering method is effective: the learned self-reflection direction aligns with an existing, model-internal axis that already separates reflective from non-reflective states, so moving along this direction amplifies or suppresses reflection by leveraging a naturally emergent subspace rather than inventing a new behavior.**

## D   IDENTIFYING SELF-REFLECTION INSTANCES

To systematically identify self-reflection in language model outputs, we developed a keyword-based detection approach. We define a self-reflection instance as any generation containing explicit self-reflection tokens that signal the model's reconsideration or revision of its reasoning process.

We construct a curated list of self-reflection keywords, informed by prior analyses of reasoning dynamics in language models (Guo et al., 2025; Liu et al., 2024b). A generation is marked as self-reflective if it contains one or more of the following terms:

To validate this detection method, we applied it to model outputs on the MATH500 benchmark using the DeepSeek-R1-Distill-Qwen-1.5B model. We set the maximum response length to 32,784 tokens to accommodate complex, multi-step solutions and to ensure that instances of late-stage self-reflection were not truncated. Among the reflection tokens, the keyword `"wait"` emerged as particularly salient. In our analysis of DeepSeek-R1 outputs, `"wait"` accounted for approximately 97.2% of all detected reflection instances. **Beyond DeepSeek-style models, `"wait"` also appears as a common reflection token in other architectures such as Llama 3.1 8B, making it a natural cross-model anchor for probing self-reflection. Importantly, our goal is not to exhaustively enumerate every**

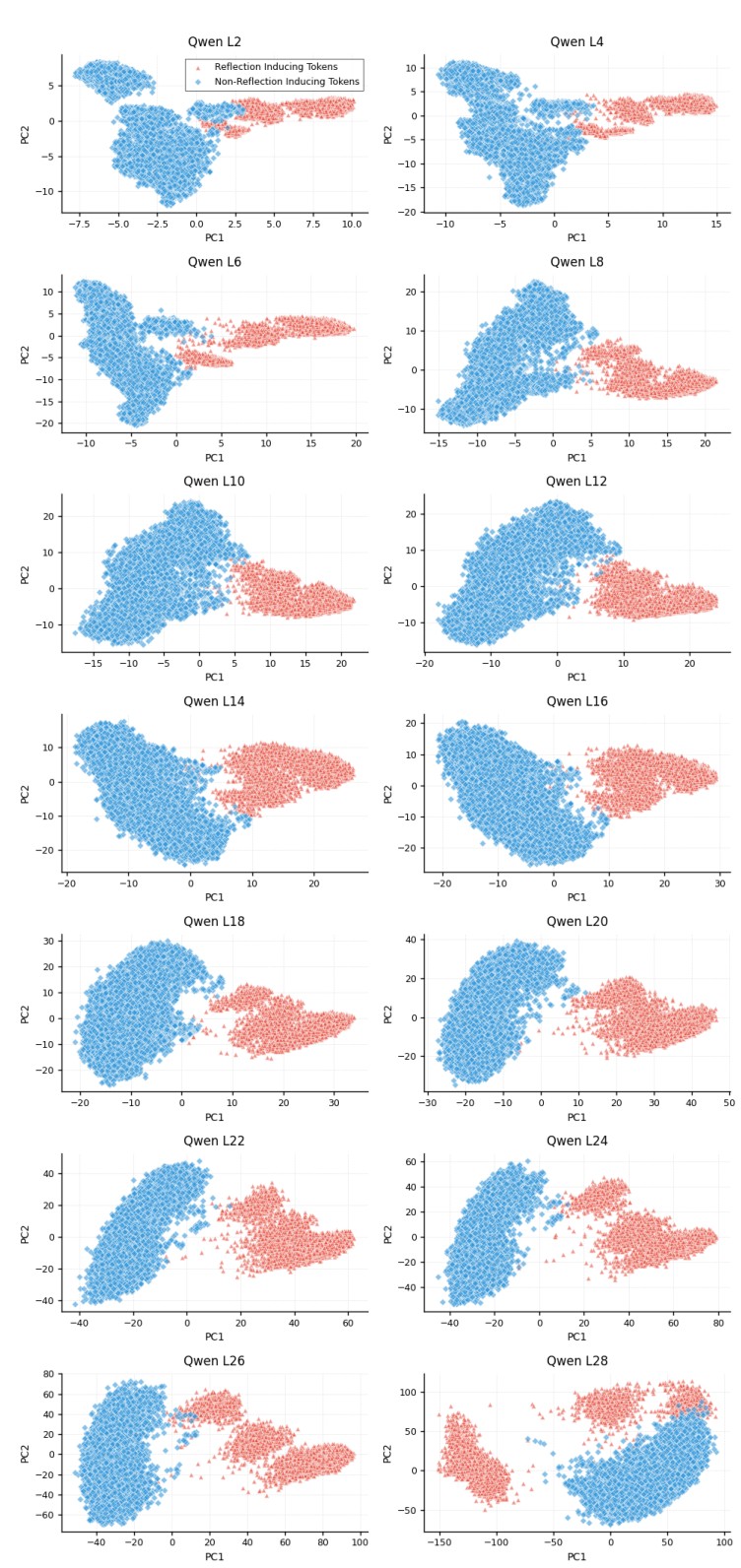

Figure 8: **PCA visualization of hidden states from the Qwen2.5-1.5B model across 14 layers.**

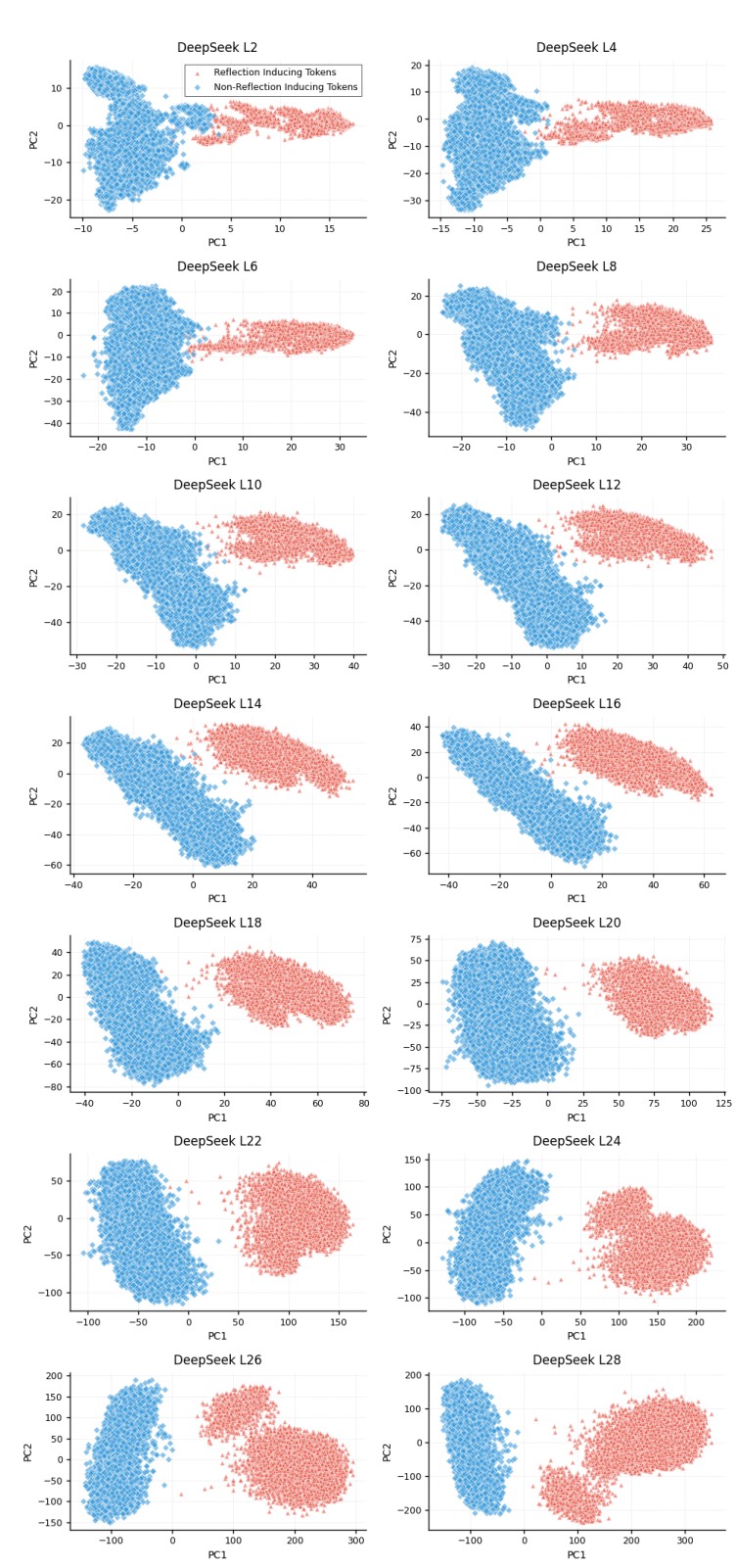

Figure 9: **PCA visualization of hidden states from the DeepSeek-R1-Distill-Qwen-1.5B model across 14 layers.**

possible reflection token, but to identify a small, high-precision subset that reliably signals self-reflective behavior. Our cross-model analysis (Section 4.6) shows that steering based on hidden states around `"wait"` nonetheless reshapes the usage of a broader family of reflection tokens and alters models' reasoning behavior, indicating that the resulting self-reflection direction is not tied to a single surface-level token pattern.

| **Dominant Self-Reflection Keyword:** |
| `wait` |

| **Miscellaneous Self-Reflection Keywords (low frequency):** |
| `re-check, recheck, check again, rethink, re-think, reconsider, re-consider, try again, re-examine, reexamine, re-evaluate, reevaluate, think again, consider again, evaluate again, examine again` |

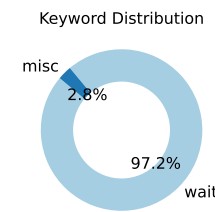

Keyword Distribution

misc 2.8%

97.2% wait

Figure 10: **Left:** Dominant and miscellaneous self-reflection keywords used in our analysis. **Right:** Frequency distribution across model-generated outputs, where `wait` constitutes the predominant share, and all other keywords occur at much lower frequencies.

# E  ABLATION STUDY

We determine the optimal injection strategy via a two-stage procedure:

(i) **Scaling Search:** For each candidate layer $\ell$, we perform a grid search over $\alpha \in [-1.0, 1.0]$ to identify the value that maximizes validation performance, exploring both enhancement and suppression regimes.

(ii) **Layer Selection:** We evaluate each layer's receptivity to injection. For specialized reasoning models (e.g., DeepSeek-R1), a single well-chosen layer often suffices to yield significant gains. In contrast, for general pretrained models (e.g., Qwen2.5), we observe that distributing moderate injections across multiple layers produces the best trade-off between accuracy and efficiency.

**Effect of Injection Layer.**  We've already described $\alpha$ selection in the main text. Here, we fix $\alpha = 0.01$ and examine the effect of injecting the self-reflection vector at each layer of DeepSeek-R1 1.5B on MATH-500 (Figure 11). We observe that middle layers, most notably layer 14, achieve the highest performance. Injections into early layers yield only marginal gains, as the steering signal is progressively transformed and attenuated by subsequent network operations. Conversely, injecting too late often degrades performance, likely because the intervention interferes directly with token generation rather than shaping deeper reasonin prg dynamics. These results indicate that moderate, mid-network interventions best modulate self-reflection by targeting layers that both abstract reasoning patterns and retain strong control over final predictions.

**Ablation study on $\alpha$**  To complement our earlier ablation on MATH500 and address the concern of evaluating only a single benchmark, we perform the same study on AIME 2024. We vary the steering strength $\alpha$ from $-1.0$ to $1.0$, injecting the reflection vector at layer 14 of DeepSeek-R1-1.5B. The results are shown in Figure 12.

Negative $\alpha$ values shorten responses substantially while preserving accuracy, confirming their utility for reducing verbosity without harming performance. Positive $\alpha$ values increase average response length and improve Pass@1, peaking around $\alpha = 0.03$ before declining as over-steering introduces excessive reflection and degrades performance.

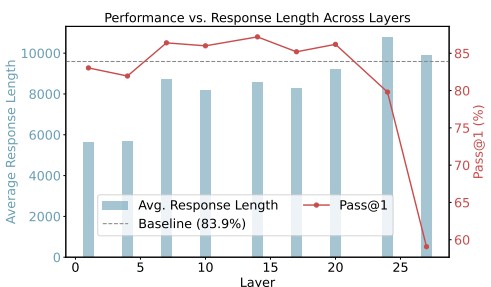

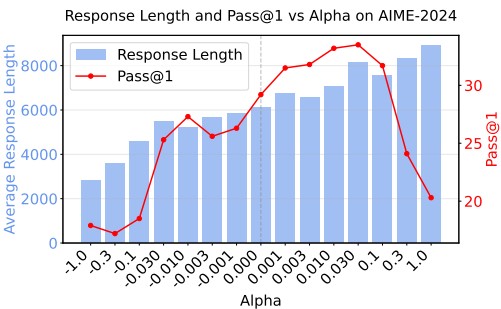

Figure 11: Effect of injecting the self-reflection vector at different layers of DeepSeek-R1 1.5B ($\alpha = 0.01$) on Pass@1 and response length for MATH-500.

Figure 12: **Effect of $\alpha$ on performance and response length on AIME 2024 (reflection vector injected at layer 14).**

## F STEERING EFFECT ON GENERAL REASONING ABILITY

To evaluate the impact of self-reflection on general ability, we report results on two metrics: faithfulness and repetition (Golovneva et al., 2023). Faithfulness assesses whether the reasoning chain misinterprets the problem or introduces vague or irrelevant information. Repetition measures redundancy between reasoning steps by computing token-level overlap. Both metrics are computed on the MATH 500 dataset, following the setup in, using `facebook/roscoe-512-roberta-base` to embed each reasoning step. As shown in Table 4, the intervention causes only negligible changes in faithfulness and repetition scores. This suggests that self-reflection does not compromise the logical coherence or diversity of the generated reasoning steps.c

Table 4: Evaluation of faithfulness and repetition on MATH500 using `roscoe-512-roberta-base` embeddings, reported for the `DeepSeek-R1-1.5B` model.

| Setting | Faithfulness ↑ | Repetition ↓ |
|---|---|---|
| Vanilla | 0.8562 | 0.0648 |
| SR Enhanced | 0.8509 | 0.0639 |
| SR Suppressed | 0.8588 | 0.0644 |

## G LIMITATIONS.

Our work presents several limitations. First, users must predefine whether to enhance or suppress self-reflection prior to inference; the model does not yet autonomously adjust its reflective behavior based on task complexity or reasoning demands. Second, our approach relies on access to internal model activations, which may not be feasible in closed-source or API-limited environments. **Third, while our most detailed layerwise and representational analyses are conducted on smaller models for computational efficiency, the broader empirical evaluation spans models from 1.5B to 13B parameters, including DeepSeek-R1 7B, Qwen2.5 7B, OLMo2-13B, and Llama 3.1 8B. Extending the full set of analyses to even larger or more heavily optimized systems remains an important direction for future work.**

In the future, these limitations could be addressed by developing adaptive self-reflection mechanisms that dynamically modulate introspection based on task complexity and reasoning signals. Further research might extend these techniques to more opaque model environments with limited activation access. Additional work could also explore methods for automatic calibration of injection parameters across diverse model architectures and reasoning domains.

