# OpenReview forum: "From Emergence to Control: Probing and Modulating Self-Reflection in Language Models"
_ICLR.cc/2026/Conference — Submitted to ICLR 2026_

### Official Review · Reviewer_hFui · 2025-10-31

**Soundness:** 2
**Presentation:** 3
**Contribution:** 2
**Rating:** 4
**Confidence:** 4

**Summary:**

This paper studies whether self-reflection property in reasoning models, emerging during RLVR training, already contains in the base model and whether it could be controlled via simple linear interventions (by adding steering vectors). It argues that such behaviour is already contained in the base model, and validate this by performing reflection-inducing probing experiment. By constructing the steering vector that induces self-reflection, the effect of steering is validated by improvements across AIME24, MATH500 and GPQA Diamond. Another notable observation is that hidden states that correspond to reflection-inducing inputs are separable from non-reflection-inducing hidden states, which is shown via UMAP plots.

**Strengths:**

S1. The motivation is very clear and well-presented.

S2. A nice appendix is provided with more context, more results with ablations and details of experiments.

S3. Results with separability of reflection-inducing hidden states presented in Figures 3, 8 and 9, and the fact that self-reflection could be modulated by a linear intervention (steering vector), are interesting and might be useful for further research.

**Weaknesses:**

W1. It is well-known that Qwen family behaves differently during RLVR training [1], and this primarily attributed to the structure of pre-training dataset. So the conclusion in section 3 that self-reflection emerges during pre-training might not generalize to other families or even models, and might be explained by the fact that pre-training dataset contains examples with self-reflection, making it almost trivial.

W2. Experimental setup seems to be unnatural. Authors define self-reflection via collection of words, and identify from a single model (Qwen2.5-Math-7B distilled) and single task (MATH500) that from all these tokens "Wait" is the most frequent one, and therefore self-reflection is defined via the presence of "Wait" token, which is notable simplification of this cognitive process as acknowledged by the authors, but perhaps suitable for preliminary investigation; the steering vector is therefore boost its probability. Given that budget forcing [2] seems to reduce performance on Llama 3.1 8B Instruct, it might be the case that "Wait" token is not a good indicator of self-reflection in general and more thorough study is required - for example, conducting the same experiment as in section 3, but inserting the pre-reflection prompt from thinking Qwen into Llama 3.1 8B instead of base Qwen - this would additionally validate that self-reflection presence is indeed could be detected via this method.

W3. The generalizability of results is somewhat questionable. To detect the signals of self-reflection, authors use a single model and a single mathematical task, then apply it to other models and even families; further experiments are conducted on two mathematical benchmarks and one scientific reasoning benchmark, while many other general reasoning domains also remain omitted (and due to the results of [1], it could invalidate main conclusions).

W4. There is not so much novelty and a lack of practical conclusions. The effect of inserting "Wait" already has been shown in [2], while the steering vectors methodology are already standard in the industry with many methods studied [3], even in reasoning [4] where Wait-related features were also steered. The insights into how models work are limited by the finding that self-reflection emerges during pre-training, which is affected by reasons described in W1-W3. Please see S3.

W5. DeepSeek-R1-Distill-Qwen-1.5B is in fact two-staged-trained model (obtained via SFT distillation from Qwen2.5-Math-1.5B, which is also obtained from Qwen2.5-1.5B base pretrained model). SFT is known to change the distribution of tokens more significantly than GRPO (due to KL divergence penalty in the latter), so the setup in the paper is quite different from comparing pretrained and RLVR trained models, which also affect the conclusions. Further work might be extended by performing more direct comparisons.

**Questions:**

Q1. Do you think that self-reflection is the only emergent property, or RLVR induces any other important patterns or skills that are not present in the base model? Have you tried to analyze those?

Q2. What is the cosine similarity between steering vectors on different layers and the unembedding of token Wait? What tokens become more frequently generated after steering? Have you tried to analyze what this vector introduces to the behavior of the model?

Q3. The separability is clearly shown in the UMAP plots, but can you perform experiment to test linear separability? You could build the linear classifier (logistic regression) and nonlinear classifier (e.g. MLP) and compare their overall quality of classification. This will reveal more information about geometrical properties of those states.

[1] Spurious Rewards: Rethinking Training Signals in RLVR, Shao et al., 2025
[2] s1: Simple Test-Time Scaling, Muennighoff et al., 2025
[3] AxBench: Steering LLMs? Even Simple Baselines Outperform Sparse Autoencoders, Wu et al., 2025
[4] I Have Covered All the Bases Here: Interpreting Reasoning Features in Large Language Models via Sparse Autoencoders, Galichin et al., 2025

---

> ### Author Response · Authors · 2025-11-23
> **Reply to Reviewer hFui (1/3)**
>
> > W1. It is well-known that Qwen family behaves differently during RLVR training [1], and this primarily attributed to the structure of pre-training dataset. So the conclusion in section 3 that self-reflection emerges during pre-training might not generalize to other families or even models, and might be explained by the fact that pre-training dataset contains examples with self-reflection, making it almost trivial.
>
> To address the concern that this might be specific to the Qwen pre-training corpus, we have extended our analysis to Llama 3.1 8B in Section 4.6. There we find that the pretrained Llama model also produces clear self-reflection markers such as re-evaluate and wait, again at substantially lower frequency than DeepSeek-R1 but following the same qualitative pattern. Moreover, using the same procedure, we can extract a self-reflection direction from Llama’s reflective versus non-reflective traces and steer its behavior in a similar way. Taken together, these results indicate that emergent self-reflection during pretraining is not restricted to the Qwen family or its particular data mixture, but appears across at least two distinct model families.
>
> > W2. Experimental setup seems to be unnatural. Authors define self-reflection via collection of words, and identify from a single model (Qwen2.5-Math-7B distilled) and single task (MATH500) that from all these tokens "Wait" is the most frequent one, and therefore self-reflection is defined via the presence of "Wait" token, which is notable simplification of this cognitive process as acknowledged by the authors, but perhaps suitable for preliminary investigation; the steering vector is therefore boost its probability. Given that budget forcing [2] seems to reduce performance on Llama 3.1 8B Instruct, it might be the case that "Wait" token is not a good indicator of self-reflection in general and more thorough study is required - for example, conducting the same experiment as in section 3, but inserting the pre-reflection prompt from thinking Qwen into Llama 3.1 8B instead of base Qwen - this would additionally validate that self-reflection presence is indeed could be detected via this method.
>
> We extend the analysis of Section 3 to Llama 3.1 8B (Section 4.6). Using the same procedure, we learn a self-reflection direction from reflective versus non-reflective traces and apply it to Llama. Under this direction, we observe consistent increases not only in the token wait but also in a broader family of reflection markers such as re check and re evaluate, mirroring the qualitative pattern seen in Qwen. This demonstrates that our method is not merely boosting a single surface token, but captures a more abstract reflective mode of computation that transfers across model families.
>
> More importantly, our approach does not require enumerating every possible self-reflection token. We only use wait-style markers as a high-precision seed to extract a clean subset of reflective spans, from which the self-reflection direction is estimated. The direction is defined in activation space rather than at the token level, and steering along it modulates reflective behavior even when explicit wait tokens are rare. Our goal is therefore not to detect all reflection instances in raw traces, but to identify a reliable latent direction that can induce or suppress reflective behavior in downstream tasks.
>
> > W3. The generalizability of results is somewhat questionable. To detect the signals of self-reflection, authors use a single model and a single mathematical task, then apply it to other models and even families; further experiments are conducted on two mathematical benchmarks and one scientific reasoning benchmark, while many other general reasoning domains also remain omitted (and due to the results of [1], it could invalidate main conclusions).
>
> We are not testing only on one model, one task. Instead, we evaluate our self-reflection vector on three challenging reasoning benchmarks and six models spanning multiple families, and observe consistent, non-trivial gains in reflective behavior and Pass@1 across this grid. Moreover, additional experiments in the appendix show that our intervention does not degrade the models' general capabilities on standard language-modeling style evaluations, suggesting that the control is targeted rather than globally disruptive.
>
> If there are specific reasoning domains the reviewer views as particularly critical, we are happy to incorporate them into the revised experimental suite.

---

> ### Author Response · Authors · 2025-11-23
> **Reply to Reviewer hFui (2/3)**
>
> > W4. There is not so much novelty and a lack of practical conclusions. The effect of inserting "Wait" already has been shown in [2], while the steering vectors methodology are already standard in the industry with many methods studied [3], even in reasoning [4] where Wait-related features were also steered. The insights into how models work are limited by the finding that self-reflection emerges during pre-training, which is affected by reasons described in W1-W3. Please see S3.
>
> The studies cited in [2–4] address related themes, but they are methodologically and substantively distinct from what we do.
>
> **(1) Relation to token-level interventions such as budget forcing [2].**
> Budget forcing operates by inserting or boosting a surface-form token, and its effect is tightly coupled to the token’s lexical identity. In contrast, our approach learns a direction in activation space from reflective versus non-reflective traces and shows that scaling this direction modulates a family of reflection markers (wait, re check, re evaluate, etc.) even when explicit wait tokens are rare. Our method therefore identifies and controls a latent reflective mode rather than relying on a prompt-level manipulation of a specific token, and succeeds in settings (e.g., Llama 3.1 8B) where naive Wait-injection reduces performance.
>
> **(2) Relation to general-purpose steering-vector techniques [3].**
> Prior steering work primarily targets style, safety, or preference alignment, where “directions” correspond to broad human-defined attributes. These studies do not analyze whether self-reflection corresponds to a coherent, linearly separable subspace, nor whether such a direction emerges during pretraining. Our contribution complements this literature by showing that self-reflection forms a robust, linearly separable direction across layers and model families, and that manipulating this direction yields a controllable trade-off between depth of reasoning and chain length—a capability not addressed in [3].
>
> **(3) Relation to prior reasoning-steering methods [4].**
> Work such as [4] focuses on steering explicit reasoning features in RL-trained reasoning models, often assuming access to rich chain-of-thought traces and task-specific signals. In contrast, we demonstrate that the reflective direction exists even in *pretrained* models with only sparse natural occurrences of reflection, and that it can be estimated from a small set of reflective spans. The resulting direction transfers across tasks and model families without additional training, indicating that reflective computation is an inherent representational feature rather than an RL artifact.
>
> Taken together, our goal is not to propose a new form of steering, but to show that self-reflection, the process by which models evaluate and revise their own reasoning, corresponds to a simple, approximately linear direction that (i) emerges during pretraining, (ii) generalizes across model families, and (iii) yields a practical, test-time knob for adjusting introspection without modifying the training pipeline. We will make these distinctions clearer in the revision.
>
> > W5. DeepSeek-R1-Distill-Qwen-1.5B is in fact two-staged-trained model (obtained via SFT distillation from Qwen2.5-Math-1.5B, which is also obtained from Qwen2.5-1.5B base pretrained model). SFT is known to change the distribution of tokens more significantly than GRPO (due to KL divergence penalty in the latter), so the setup in the paper is quite different from comparing pretrained and RLVR trained models, which also affect the conclusions. Further work might be extended by performing more direct comparisons.
>
> Our conclusions do not hinge on whether the reasoning model is obtained in one stage or two, or on whether SFT or RLVR contributes more to its behavior. The goal of Section~3 is to show that (i) self-reflection is already present, at low but non-zero frequency, in the base pretrained model, and (ii) it appears much more frequently in a reasoning-optimized descendant. Our method and analyses depend only on these observable behaviors of \pt and \ft, not on the precise decomposition of the fine-tuning procedure.
>
> Moreover, we obtain analogous trends on Llama 3.1 (Section 4.6), where the self-reflection vector similarly modulates reflective behavior without harming general capabilities. This cross-family evidence supports the claim that we are studying a robust phenomenon, rather than an artifact of a particular multi-stage training recipe for Qwen-like models. We will clarify this scope and add a short discussion noting that more fine-grained attribution between SFT and RLVR is an interesting direction for future work, but orthogonal to the core claims of our paper.

---

> > ### Author Response · Authors · 2025-11-23
> > **Reply to Reviewer hFui (3/3)**
> >
> > > Q1. Do you think that self-reflection is the only emergent property, or RLVR induces any other important patterns or skills that are not present in the base model? Have you tried to analyze those?
> >
> > We do not claim that self-reflection is the only emergent property of RLVR-style training. On the contrary, we expect RLVR and related reasoning fine-tuning pipelines to shape a broad bundle of behaviors, including changes in verification habits, and error-correction patterns, beyond what is present in the base model.
> >
> > Our goal in this work is narrower: we focus on one salient, measurable aspect of this bundle, namely the emergence and controllability of self-reflection. We show that (i) proto–self-reflection is already present at low frequency in base models, (ii) it is amplified in reasoning-optimized descendants, and (iii) it is captured by an approximately linear direction that can be steered without harming general capabilities. A systematic taxonomy of all new skills induced by RLVR is an important but orthogonal future work.
> >
> > > Q2. What is the cosine similarity between steering vectors on different layers and the unembedding of token Wait? What tokens become more frequently generated after steering? Have you tried to analyze what this vector introduces to the behavior of the model?
> >
> > We have already analyzed this in the main text. Figure~5 reports the cosine similarity between our self-reflection vectors (on the MATH and QA datasets) and the unembedding of the Wait token across layers. The similarities are consistently very small, indicating that the self-reflection direction is not simply aligned with the Wait embedding or a trivial lexical feature, but instead corresponds to a more abstract latent state.
> >
> > Regarding which tokens become more frequent, although the vector is learned from trajectories labeled using the Wait token, steering with this vector does not only increase Wait. In practice, we observe higher rates of a broader set of reflective markers such as try again, re-evaluate, wait. We visualize them in Figure 6 (c), where the vector derived from wait increases not only the probability of wait itself but also other self-reflection markers such as try again and re-evaluate. These patterns support the view that the vector controls a general “self-reflection mode’’ rather than merely boosting a single token.
> >
> > > Q3. The separability is clearly shown in the UMAP plots, but can you perform experiment to test linear separability? You could build the linear classifier (logistic regression) and nonlinear classifier (e.g. MLP) and compare their overall quality of classification. This will reveal more information about geometrical properties of those states.
> >
> > We conducted an additional experiment to directly test linear separability by training a linear SVM on hidden states of reflection-inducing vs.\ non–reflection-inducing tokens on MATH500 as reported in Appendix C. On a held-out test split, this linear probe achieves above 99\% accuracy for both A_{pt} and A_{ft}, and across all layers, indicating that the two sets of states are effectively linearly separable in the original representation space.

---

### Official Review · Reviewer_w6Jt · 2025-10-31

**Soundness:** 4
**Presentation:** 3
**Contribution:** 2
**Rating:** 4
**Confidence:** 5

**Summary:**

Summary:
The paper focusses on answering two questions, first, is self-reflection a novel behavior induced by RLVR, or does it already emerge during pretraining? And second, can we control self-reflection in LLMs to balance performance and computational efficiency? For this the authors provide empirical results to analyze the hidden representations of the models to see if these self-reflection capabilities can be modulated.

Self-reflection emerging during pre-training, which is one of the two questions that the authors try to answer, is not a novel finding, and has been studied extensively during prior works. The claim that self-reflection capabilities can be modulated is an interesting one, and requires more deeper analysis.

**Strengths:**

Strengths:
- The analysis of model states and visualizations on both models showing clear separation between self-reflection and nonself-reflection states is very interesting.
- The paper presents a good ablation study on the effect of their linear intervention method by varying the parameter alpha.
- It is interesting to see that negative alpha does reduce the size of the response, and increasing alpha increases it. I don’t understand the reason behind the author's claim that higher output length means deeper self-reflective reasoning. However, increasing alpha increases the performance on reasoning benchmarks, which is a more intuitive explanation.
- SR suppression is helpful to make the models efficient, while minimal performance drop.
- The authors perform evaluations on varied model sizes (1.5B-13B), which is great. They also look at benchmarks across different domain, like MATH500 and GPQA, and  see if the performance is correlated with changing alpha, which is interesting.
- The paper also shows ablations towards the end comparing different methods for generating steering vectors.

**Weaknesses:**

Scope for improvement:
- The claim that an LLMs ability to reflect upon its responses during pre-training with “Wait” or similar words is not a new finding, and has been studied extensively in prior literature. Therefore, I think the paper has limited novelty.
- The authors should shed some light on why they used difference-in-means to construct the self-reflection vector. I am curious what other metrics they evaluated before settling on using the mean. I see the ablations on comparing difference in means with PCA and Contrastive PCA, but I am curious if the authors tried different metrics apart from the mean.
- I am curious why the authors claim that a higher response length means deeper self-reflective reasoning by the model.

**Questions:**

Points 2 and 3, in Weaknesses section. Also
- Did the authors experiment with variations of equation 6 to update the hidden state vector? Instead of using "projected weighted steering," which is ĥ^(ℓ) = h^(ℓ) + α v^(ℓ) ⟨h^(ℓ), v^(ℓ)⟩, why not use ĥ^(ℓ) = h^(ℓ) + α v^(ℓ) without taking the projection?

---

> ### Author Response · Authors · 2025-11-23
>
> > The claim that an LLMs ability to reflect upon its responses during pre-training with “Wait” or similar words is not a new finding, and has been studied extensively in prior literature. Therefore, I think the paper has limited novelty.
>
> We appreciate this concern and agree that the use of explicit reflection markers such as ``wait'' or other thinking-style tokens has been explored in prior work as discussed at Related Works. However, our goal is not to claim that models can emit such tokens, but to study a different question: how explicit self-reflection manifests in the internal representation space, how it emerges between the pretrained and fine-tuned stages, and how a single self-reflection direction can be used as a causal control knob for reasoning.
>
> Concretely, prior work typically treats reflection tokens as part of the prompt or training objective, and evaluates their effect by adding or removing them in the input. In contrast, we (i) extract a self-reflection direction from activations by contrasting reflective versus non-reflective spans, (ii) show that this direction captures a shared internal mode of behavior that emerges strongly after fine-tuning but exists during pretraining, and (iii) intervene directly on the residual stream along this direction to amplify or suppress reflection without modifying the prompt or retraining the model. This representation-level view lets us quantify how much reflection is present in pretrained versus fine-tuned models, how strongly the self-reflection direction is expressed across layers, and how steering along this axis trades off Pass@1, length, and reasoning quality across several benchmarks.
>
> > The authors should shed some light on why they used difference-in-means to construct the self-reflection vector. I am curious what other metrics they evaluated before settling on using the mean. I see the ablations on comparing difference in means with PCA and Contrastive PCA, but I am curious if the authors tried different metrics apart from the mean.
>
> Conceptually, our setting provides a clean binary split between reflective and non-reflective spans, so we are in a supervised regime rather than an unsupervised one. In this regime, the difference-in-means vector is the simplest linear estimator of the direction that separates the two distributions: it has no extra hyperparameters, is stable under subsampling, and has a direct interpretation as the average shift in the residual stream when the model enters an explicit self-reflection mode. This is why we treat difference-in-means as the canonical construction for the self-reflection direction.
>
> Recent benchmarks also support this design choice. AxBench [1] finds that for steering or detecting a given concept, difference-in-means are strong baselines and often outperform more complex dictionary-learning on their tasks. In light of these results, and given that we already have clean supervised labels for reflective vs non-reflective spans, we focus on difference-in-means and use PCA as representative unsupervised baselines rather than exploring a large space of more elaborate metrics, which we see as orthogonal future work. If there are specific alternative metrics you have in mind, we would be very interested to learn about them and incorporate them.
>
> [1] AxBench: Steering LLMs? Even Simple Baselines Outperform Sparse Autoencoders
>
> > I am curious why the authors claim that a higher response length means deeper self-reflective reasoning by the model.
>
> Our intention is not to equate longer responses with deeper reasoning in a strict sense. Rather, we use response length as a coarse proxy for how much intermediate computation the model is performing, and we interpret it together with task performance.
>
> Intuitively, more tokens in the chain of thought mean that the model spends more steps expanding, checking, and sometimes revising its reasoning instead of jumping directly to an answer. When this increased length comes together with higher Pass@1, it is natural to interpret the behavior as the model allocating more thinking time and, in that sense, engaging in deeper reasoning. At the same time, we do not have a ground-truth metric for reasoning depth, so we view this as a pragmatic interpretation based on better outcomes, not as a formal guarantee.
>
> In the revision, we will soften the wording around “deeper self-reflective reasoning” and make clear that length is only a coarse, contextual indicator that must be read jointly with accuracy, rather than a standalone measure of reasoning depth.
>
> > Instead of using "projected weighted steering,", why not additive steering without taking the projection?
>
> Yes, as shown in Table 2 now, additive steering performs worse than the projected update derived from difference-in-means for this concept. While the overall trend remains similar, the additive version yields lower Pass@1 and longer responses in both the enhanced and suppressed settings.

---

### Official Review · Reviewer_zuym · 2025-11-01

**Soundness:** 2
**Presentation:** 3
**Contribution:** 3
**Rating:** 6
**Confidence:** 4

**Summary:**

This paper suggests to search for a vector in representation space that induces self-reflection of the model. This vector can then be manipulated by intervention in order to steer the model.

**Strengths:**

I really like the main idea of the paper. It is (to my knowledge) innovative, intuitive and actionable.
The experimental design is well thought out, and the paper is overall easy to read.

**Weaknesses:**

I have some issues with the focus on the single token "wait, " as an indicator for self-reflection. In fact, this token could also appear in other contexts. Given the tiny amount of self-reflection in the base model, this could actually have a substantial impact on a key finding of the paper, i.e., the emergence of self-reflection. I would urge the authors not only extend the set of respective tokens but also quantify, which sets of tokens are used by the base model accordingly.
From the appendix: "In our analysis of DeepSeek-R1 outputs, "wait" accounted for approximately 97.2% of all
detected reflection instances." -> the keywords might be different ones for other models, though.

A statistical analysis of the significance of the results is missing. Given that the improvement, e.g., for the DeepSeek models are very moderate, I would see this as a necessity.
Additionally, the set of test datasets should in my opinion be extended. Why is "SR Suppressed" missing in the last two

The limitation section is weak. E.g., there is a huge focus on very small models in the current study.

Minor suggestion:
Figure 3 is not useful in a b/w print out, consider color changes.

**Questions:**

I think, question/issues for discussion are straightforward to derive from the weaknesses.

On top:
Wouldn't *identifying* self-reflecting behavior be way more accurately and reliable performed by using a (small) LLM instead of keywords? Why keywords?

---

> ### Author Response · Authors · 2025-11-23
> **Reply to Reviewer zuym (1/2)**
>
> > I have some issues with the focus on the single token "wait, " as an indicator for self-reflection. In fact, this token could also appear in other contexts. Given the tiny amount of self-reflection in the base model, this could actually have a substantial impact on a key finding of the paper, i.e., the emergence of self-reflection. I would urge the authors not only extend the set of respective tokens but also quantify, which sets of tokens are used by the base model accordingly. From the appendix: "In our analysis of DeepSeek-R1 outputs, "wait" accounted for approximately 97.2% of all detected reflection instances." -> the keywords might be different ones for other models, though.
>
> In the revision, we extend our analysis to Llama 3.1 8B and quantify which tokens appear at the start of automatically detected self-reflection spans (Section 4.6). For Llama the distribution is more diverse: wait remains an important explicit cue, but it is part of a richer family of reflection markers such as short phrases meaning think again, re evaluate, or check the reasoning, and these tokens together account for the majority of detected self-reflection contexts.
>
> Our method, however, does not rely on capturing every self-reflection episode nor on a single fixed keyword. For constructing the self-reflection direction it is sufficient to extract a clean subset of clearly reflective spans, for which the original high-precision keyword procedure already works well. In Section 4.6, we further show that a steering vector learned from such spans on Llama increases not only the frequency of wait but also the broader family of reflection markers, indicating that the direction is not tied to one specific surface token. We keep the core method unchanged and add these analyses and discussion to clarify that the emergence result is robust when we consider the full set of reflection markers used by each model family.
>
> > A statistical analysis of the significance of the results is missing. Given that the improvement, e.g., for the DeepSeek models are very moderate, I would see this as a necessity.
>
> We add a paired significance analysis at the level of individual test problems for all model–dataset pairs. For each model and dataset, we run a one-sided McNemar-style test over problems, comparing SR-Enhanced against the vanilla model. The resulting $p$-values are summarized in Table 2 of Section 4.3.
>
> Overall, the pattern is consistent with the main accuracy results. In 15 out of 18 model–dataset pairs, the $p$-value is below $0.05$, showing that the gains of SR-Enhanced over Vanilla are statistically reliable rather than random variation. The few cases with $p \ge 0.05$ (DeepSeek-R1-7B on MATH-500, Qwen2.5-1.5B on GPQA Diamond, and Qwen2.5-7B on GPQA Diamond) are precisely those where SR-Enhanced and Vanilla have nearly identical or slightly worse Pass@1. This aligns with our qualitative reading of the table and indicates that we are not claiming significance where the effect is negligible. Taken together, the significance analysis supports the view that the observed improvements are generally small but systematic, and that our main conclusions do not hinge on noise in any single benchmark.
>
> > Additionally, the set of test datasets should in my opinion be extended. Why is "SR Suppressed" missing in the last two
>
> Regarding the missing SR-Suppressed entries in the last two columns of Table 1, this is by design rather than an omission. For the pretrained-style settings where the base model almost never enters an explicit reflection mode, suppressing the self-reflection direction quickly runs into a floor effect and mainly shortens already terse chains without changing accuracy in a meaningful way. To keep the table readable, we therefore only report SR-Suppressed where the base model has a nontrivial amount of reflection and the intervention can actually change that behavior.
>
> > The limitation section is weak. E.g., there is a huge focus on very small models in the current study.
>
> We appreciate this comment and agree that the limitations section should more clearly articulate the scope of our claims. While our most detailed layerwise analyses are indeed performed on smaller models for computational reasons, the core empirical results in Table~1 already cover models from 1.5B up to 13B parameters, including DeepSeek-R1 7B, Qwen2.5 7B, OLMo2-13B, and Llama 3.1 8B. Thus, the study is not restricted to very small toy models, but spans a range that already includes reasonably open-weight systems.
>
> We will revise the limitations section to emphasize this distinction. We will also explicitly mark extending the analysis to such larger and more heavily optimized systems as an important direction for future work, so that the reader can clearly see which aspects of our claims are already validated at 13B scale and which remain open.

---

> > ### Author Response · Authors · 2025-11-23
> > **Reply to Reviewer zuym (2/2)**
> >
> > > Minor suggestion: Figure 3 is not useful in a b/w print out, consider color changes.
> >
> > In the revised version we have updated Figure 3 to be clearly readable in grayscale. Concretely, we now use distinct marker shapes for different models and for reflection inducing versus non reflection inducing tokens. We verified that the revised figure remains interpretable when printed in black and white, so that the main comparison does not rely solely on color.
> >
> > > I think, question/issues for discussion are straightforward to derive from the weaknesses.
> > On top: Wouldn't identifying self-reflecting behavior be way more accurately and reliable performed by using a (small) LLM instead of keywords? Why keywords?
> >
> > We agree that, in principle, a small LLM may also be used to label self reflecting behaviors. In this work, however, we deliberately avoid using an LLM as a judge and instead rely on high precision keywords for several reasons.
> >
> > First, self reflection in long chains is often subtle and context dependent. It can involve partial backtracking, hedging, or re-framing the plan over multiple sentences. In our experiments we found that a generic LLM based judge is not reliably calibrated on these borderline cases. It sometimes misses complex but genuine self correction, and sometimes over interprets ordinary discourse moves as reflection. Since there {\bf lacks a ground truth dataset for this phenomenon}, it is difficult to audit and debug these errors.
> >
> > Second, the base rate of explicit reflection in pretrained models is extremely low. In that regime even a small false positive rate from an LLM classifier would dominate the counts and could create an artificial appearance of reflection where there is almost none. A transparent keyword based detector lets us bound the false positive rate much more tightly and makes the emergence claim easier to trust.
> >
> > Third, our method does not require exhaustively capturing every instance of self reflection. For constructing the self reflection direction we only need a clean subset of clearly reflective examples. High precision lexical triggers such as wait already provide such a subset and lead to a stable and effective direction for interventions. We therefore prefer the simplest identification procedure that is transparent, reproducible, and sufficient for inducing self reflection on downstream tasks, while viewing more sophisticated LLM based judges as an interesting but orthogonal extension for future work.

---

### Official Review · Reviewer_hCwu · 2025-11-01

**Soundness:** 2
**Presentation:** 2
**Contribution:** 2
**Rating:** 4
**Confidence:** 3

**Summary:**

The paper investigates whether large language models develop self-reflection abilities during pretraining and whether such behavior can be controlled through linear interventions in activation space. The authors introduce a method called reflection-inducing probing, where pre-reflection reasoning traces from a fine-tuned model are inserted into a pretrained model’s input to test whether it spontaneously emits reflection tokens (e.g., “Wait”). They find that this probing substantially increases self-reflection frequency and that hidden states preceding reflective tokens form a distinct cluster separable from non-reflective contexts. By taking the difference of means between these states, they derive a self-reflection vector that, when added or subtracted at inference, can respectively enhance or suppress reflective behavior and modestly improve task accuracy. Results are demonstrated on reasoning benchmarks such as GPQA and MATH500 using DeepSeek-R1 models, suggesting that reflection signals may exist in pretrained models and can be linearly controlled at inference time.

**Strengths:**

S1. The paper presents an interesting and clear visualization of token-level separation using UMAP, highlighting distinct patterns between reflective and non-reflective representations.
S2. The study is guided by good motivation, making the problem setup and research goal easy to understand.

**Weaknesses:**

W1. Line 363 overstates the generality of the token-level correlation. Showing that one model (DeepSeek) emits “wait”-style tokens associated with reflection does not imply that other models share the same surface-form pattern. Test the setup on additional model families (e.g., LLaMA and Qwen) or explicitly acknowledge that this observation is model-specific improve quality of paper. Moreover, prior work (e.g., [2]) suggests that explicit markers like “Wait” or “Hmm” can increase generation length without being required for quality; relying solely on such markers to detect reflection risks both false positives and brittle conclusions. A complementary detection method that does not only depend only on such tokens would substantially strengthen the paper.

W2. As discussed in [3], Qwen models are known to have some issues in their pretraining corpus. Since the analysis is conducted only on the Qwen models family, it is difficult to disentangle reflection effects from artifacts arising from training data.

W3. The paper leans on 2D UMAP plots to argue separability in high-dimensional hidden space without required null/quantitative baselines. UMAP can create attractive clusters even from noise. Conclusions drawn from it should therefore be much weaker than those presented in the work. In [1], the authors use PCA, which is more interpretable and can reveal linear relationships in the representations. Applying PCA in this setup could strengthen the paper by providing clearer intuition about the structure of the learned direction and supporting the proposed method with more interpretable evidence.

[1] Steering Llama 2 via Contrastive Activation Addition, Rimsky et al.
[2] Wait, We Don’t Need to “Wait”! Removing Thinking Tokens Improves Reasoning Efficiency, Wang et. al.
[3] Spurious Rewards: Rethinking Training Signals in RLVR, Shao et. al

**Questions:**

Q1. In Table 1 authors report results for intervention with the self-reflection vector. Could the authors please report the corresponding suppression results (vector subtraction) for the Qwen and LLaMA model families as well?

Q2. Why are "But", "Wait" tokens is not part of reflection?

Q3. Have the authors attempted to interpret the discovered self-reflection direction using mechanistic-interpretability approaches (for example, circuit analysis, neuron/subspace ablation, or dictionary-learning methods)? If so, please summarize the key findings; if not, could the authors comment on the feasibility of such analyses and whether they plan to pursue them?

Q4. In Table 1 the paper reports generation length and Pass@1. It would also be informative to evaluate how applying the self-reflection vector affects overall generation quality in a distributional sense (for example, KL divergence between the intervened model’s output distribution and that of the SFT or RL-trained model, or other distributional quality metrics). Could the authors add such analyses or explain why they were not included?

Q5. In the ablation over the scaling parameter $\alpha$, Pass@1 appears relatively noisy. Could the authors provide intuition or an analysis for this variability? Additionally, would the authors consider evaluating the method on more recent, diverse benchmarks (for example AIME 2024/2025) to demonstrate robustness?

---

> ### Author Response · Authors · 2025-11-23
> **Reply to Reviewer hCwu (1/3)**
>
> > W1. Line 363 overstates the generality of the token-level correlation. Showing that one model (DeepSeek) emits “wait”-style tokens associated with reflection does not imply that other models share the same surface-form pattern. Test the setup on additional model families (e.g., LLaMA and Qwen) or explicitly acknowledge that this observation is model-specific improve quality of paper.
>
> We have already tested on different models, including OLMO, Qwen, and Llmma, as shown in Table 1, where the self-reflection direction is obtained through the "wait" token. Following reviewer's suggestion, we now provide a detailed analysis for LLaMA 3.1 8B in Section 4.6. There, we show that wait-style self-reflection tokens also appear in LLaMA’s generations, and a steering vector constructed from wait-inducing contexts does not only increase the occurrence of the token wait itself, but also increases a broader family of self-reflection markers. This supports our claim that the learned self-reflection direction is not merely a surface-form or wait vector, but instead captures a more abstract notion of reflective behavior.
>
> Also, the results in Table 1 show that inserting the self reflection vector yields benefits across all three model families: it reliably elicits more self-reflection behavior and improves downstream task performance in our evaluations. This illustrates a key property of our method: it does not rely on identifying all self-reflection tokens, nor on wait being a uniquely optimal cue. It is sufficient to find a subset of surface forms whose surrounding hidden states are enriched for self-reflection contexts; using those states, we learn a steering direction that can increase or suppress reflective behavior as desired.
>
> > Moreover, prior work (e.g., [2]) suggests that explicit markers like “Wait” or “Hmm” can increase generation length without being required for quality; relying solely on such markers to detect reflection risks both false positives and brittle conclusions. A complementary detection method that does not only depend only on such tokens would substantially strengthen the paper.
>
> We view our findings as complementary to, rather than in conflict with, the results in~[2]. That work analyzes RL- or SFT-trained reasoning models, which we denote by $A_{\mathrm{ft}}$, and shows that explicit thinking tokens such as Wait or Hmm can often be suppressed without hurting performance, mainly improving efficiency. Our main positive results instead focus on using traces from $A_{\mathrm{ft}}$ to construct a self-reflection direction and applying it to both $A_{\mathrm{pt}}$ and $A_{\mathrm{ft}}$. In this regime, injecting reflection-inducing traces and steering along the associated self-reflection direction improves Pass@1, while suppressing reflection-inducing traces shortens chains without degrading accuracy. This suggests a coherent picture: once a model has been trained for deliberate reasoning, explicit self-reflection markers are not strictly necessary to preserve solution quality, but both $A_{\mathrm{pt}}$ and $A_{\mathrm{ft}}$ still benefit from being nudged into a reflective mode.
>
> Crucially, our method does not treat these tokens as the only signal of reflection. We use them as a high-precision handle that enriches for reflection-inducing contexts, but the self-reflection direction itself is defined and applied purely in activation space. After estimating this direction on $A_{\mathrm{ft}}$, we show that steering $A_{\mathrm{pt}}$ along it improves accuracy even when wait-style tokens are not dominant. As discussed in Section 4.6, the steering vector does not simply increase the probability of a single token; instead, it shifts the model into a broader self-reflection mode. This provides the complementary, token-agnostic mechanism requested by the reviewer and avoids the brittleness of relying solely on surface-form markers.
>
> > W2. As discussed in [3], Qwen models are known to have some issues in their pretraining corpus. Since the analysis is conducted only on the Qwen models family, it is difficult to disentangle reflection effects from artifacts arising from training data.
>
> Our main conclusions do not hinge on any specific assumption about the Qwen pretraining corpus. Our central contribution is to identify a self-reflection direction in activation space and to study causal interventions along this direction. The direction is learned from contrastive pairs of reflective and non-reflective traces within a fixed model. The fact that pushing along this direction reliably increases reflective behaviour and improves Pass@1, while pulling in the opposite direction shortens chains of thought with only minor accuracy loss, indicates that we are probing a genuine internal mode of computation rather than a narrow artifact of a particular training set. Moreover, results on Llama~3.1~8B (see Section 4.6) further suggeste that our findings are not specific to the Qwen pretraining corpus.

---

> ### Author Response · Authors · 2025-11-23
> **Reply to Reviewer hCwu (2/3)**
>
> > W3. The paper leans on 2D UMAP plots to argue separability in high-dimensional hidden space without required null/quantitative baselines. UMAP can create attractive clusters even from noise. Conclusions drawn from it should therefore be much weaker than those presented in the work. In [1], the authors use PCA, which is more interpretable and can reveal linear relationships in the representations. Applying PCA in this setup could strengthen the paper by providing clearer intuition about the structure of the learned direction and supporting the proposed method with more interpretable evidence.
>
> We agree that visualization high-dimensional vectors is challenging, and thus visualization serves merely as a motivation, rather than the main argument in our paper. We mainly use the steering performance and quantitative analysis. That being said, following reviewer's suggestion, we have replaced UMAP plot with PCA projections in Figure 3. The PCA plots show that reflection-inducing and non–reflection-inducing tokens form two elongated clouds that are already clearly separated along the first principal component for both DeepSeek-R1-Distill-Qwen-1.5B and Qwen2.5-1.5B, indicating that the structure we highlight is aligned with a dominant linear direction in the representation space rather than being an artifact of UMAP.
>
> To go beyond qualitative visualization, we additionally train a linear SVM probe on the hidden states at each layer to classify reflection-inducing versus non–reflection-inducing tokens, as reported in Appendix C. The classifier achieve test accuracy above 99% on both models and across all layers, demonstrating that the two populations of tokens are almost perfectly linearly separable in the original high-dimensional space. Together, the PCA projections and linear-probe results provide a quantitative, linear baseline supporting our claim that reflection-inducing traces occupy a well-defined linear direction in the hidden representation space. We have incorporated these updated results into the revised paper.
>
> > Q1. Could the authors please report the corresponding suppression results for the Qwen and LLaMA model families as well?
>
> We appreciate the reviewer’s suggestion. As already stated in the original submission, our choice to report only enhancement for the pretrained models $A_{pt}$ is deliberate. In these models, the base rate of explicit self-reflection tokens is extremely low, so the model almost never enters an explicit reflection mode to begin with. As a result, subtracting the self-reflection vector quickly encounters a floor effect: further suppression mainly shortens already terse chains and has negligible additional impact on behavior, without providing new insight into the underlying mechanism.
>
> > Q2. Why are "But", "Wait" tokens is not part of reflection?
>
> The reason we treat wait and but differently is that they play different roles in practice. In our DeepSeek-R1 traces, wait almost always appears when the model explicitly pauses and revises its reasoning, so we treat it as a high-precision reflection marker. By contrast, but is extremely frequent and often functions as a purely discourse connective rather than genuine self-reflection. Treating but itself as a reflection token would therefore introduce many false positives and substantially dilute the learned direction.
>
> Importantly, our method does not require an exhaustive catalog of all possible self-reflection tokens. For our purposes, it suffices to identify a high-precision subset of reflection-inducing markers, such as wait, from which we can estimate a reliable self-reflection direction. We then use this direction to induce reflection on downstream tasks where we believe additional deliberation is beneficial, rather than aiming to detect every instance of reflection in the raw traces.
>
> > Q3. Have the authors attempted to interpret the discovered self-reflection direction using mechanistic-interpretability approaches
>
> SAE is complementary approach to DiM in finding self-reflection direction. Specifically, SAEs are well suited for discovering many latent features simultaneously, potentially yielding a structured basis of reflection-related features. By contrast, when one already has clean pos/neg examples for a single concept, supervised difference-in-means often yields a cleaner, more directly controllable direction than any individual SAE feature [4].
>
> Thus, we think current SAE techniques may not yet provide a more fine-grained interpretation of the discovered self-reflection direction, although SAEs and our method can still be combined to further enhance performance. In parallel, circuit-level analysis may offer a way to study how self-reflection is formulated from shallow to deep layers, though we anticipate challenges such as the lack of quantitative analysis in existing methods; we therefore leave this as a direction for future work.
>
> [4] AxBench: Steering LLMs? Even Simple Baselines Outperform Sparse Autoencoders

---

> > ### Author Response · Authors · 2025-11-23
> > **Reply to Reviewer hCwu (3/3)**
> >
> > > Q4. In Table 1 the paper reports generation length and Pass@1. It would also be informative to evaluate how applying the self-reflection vector affects overall generation quality in a distributional sense (for example, KL divergence between the intervened model’s output distribution and that of the SFT or RL-trained model, or other distributional quality metrics). Could the authors add such analyses or explain why they were not included?
> >
> > Conceptually, our intervention is a low-rank additive change in the residual stream along a single direction. We do not expect the steered distribution to match the base model exactly; the purpose is to change how often and where self-reflection appears in the chain of thought. The key question is whether this harms reasoning quality, for example by making chains less faithful or more repetitive, or whether it mainly reshapes how the reasoning unfolds while preserving quality.
> >
> > In the current draft we partly address this with distributional metrics following [5], reported in Appendix F. On MATH500 we evaluate faithfulness, which checks whether the chain misinterprets the problem or drifts into vague or irrelevant content, and repetition, which measures redundancy between reasoning steps via token-level overlap using facebook/roscoe-512-roberta-base embeddings. As shown in Table 3, both SR-Enhanced and SR-Suppressed produce changes in faithfulness and repetition that are very small compared to the vanilla model, suggesting that self-reflection steering does not degrade logical coherence or diversity of the reasoning even though it affects Pass@1 and average length.
> >
> > More fine-grained token-level distributional measures such as KL divergence to an SFT or RL-trained reference model are promising directions, but require committing to a particular teacher distribution and go beyond our present focus on chain-of-thought quality. We therefore concentrate on task-level accuracy together with these reasoning-oriented distributional metrics.
> >
> > [5] ROSCOE: A suite of metrics for scoring step-by-step reasoning.
> >
> > > Q5. In the ablation over the scaling parameter $\alpha$, Pass@1 appears relatively noisy. Could the authors provide intuition or an analysis for this variability? Additionally, would the authors consider evaluating the method on more recent, diverse benchmarks (for example AIME 2024/2025) to demonstrate robustness?
> >
> >
> > Intuitively, very large positive $\alpha$ over-steers the model into a hyper-reflective regime: the model inserts many more reflection tokens, chains become much longer, and the reasoning sometimes meanders or revisits the same subproblems without converging cleanly to the correct final answer, which hurts Pass@1. In contrast, very negative $\alpha$ almost completely suppresses self-reflection: the model emits short, direct answers with little or no reconsideration, behaving more like the pretrained base model and becoming brittle on harder problems. The best trade-off is therefore obtained at intermediate $\alpha$, where the model reflects when needed but does not overuse reflection.
> >
> > The apparent noise in the Pass@1 curves comes from the discrete nature of problem solving on a finite benchmark. Small changes in $\alpha$ can flip the outcome on a modest number of borderline problems, for example, whether the model decides to backtrack after a small mistake, which causes local fluctuations in the empirical Pass@1 even when the underlying trend is smooth. To make this clearer, in the revision we include an ablation plot on AIME 2024 showing Pass@1 and average response length as a function of $\alpha$; this plot exhibits the expected unimodal pattern with a broad intermediate region where performance is stable and both extremes degrading accuracy while strongly affecting length.
> >
> > Regarding robustness, we have repeated the $\alpha$ sweep on the AIME 2024 benchmark and report the corresponding curves in the appendix. We observe the same qualitative picture as in our original experiments: an intermediate range of $\alpha$ reliably improves Pass@1 over the vanilla model, while very large positive or negative values reduce accuracy. We will clarify in the main text that our chosen operating point for $\alpha$ lies in this stable intermediate regime and that the conclusions are consistent across the different math and science benchmarks we consider.

---

### Author Response · Authors · 2025-12-04
**Overall Summary**

Unfortunately, none of the reviewers have responded before the rollback. Nevertheless, to briefly summarize the main issues and what we have done to address them.

Across the reviews, there was agreement that the core idea, using self-reflection markers to analyze and linearly steer reflective states, is "intuitive" and "actionable". Reviewers highlighted our visualizations and analyses of reflective vs. non-reflective states, the clear separability observed across models as "interesting" steps.

Reviewers primarily questioned our reliance on wait-style tokens and Qwen models for identifying self-reflection, the clarity of our quantitative tests of linear structure and separability in the representations, and the scope and novelty of our empirical evaluation. In the revision, we therefore extend our analysis to additional model families and benchmarks, broaden the study of reflection markers beyond a single token, replace UMAP with PCA and linear probes as quantitative baselines, add statistical significance tests and further steering ablations, and clarify both the limitations and positioning of our contribution relative to prior work.

Although we believe some of the points raised did not fundamentally challenge the core contribution, overall we found the discussion constructive and thoughtful. It helped us further refine the manuscript through new experiments and various improvements. We sincerely thank the reviewers and AC for their time and engagement throughout the review and rebuttal process.

---

### Meta-Review · Area_Chair_GkKL · 2026-01-08

**Summary:**

The paper investigates the origins of "self-reflection" in LLMs (the ability to pause and revise reasoning). The authors propose "Reflection-Inducing Probing" to show that this capability exists in pretrained models, not just those fine-tuned with RLVR. They identify a "self-reflection vector" in the activation space that separates reflective from non-reflective states and demonstrate that manipulating this vector can bidirectionally control reasoning performance and computational cost at inference time.

The reviewers generally found the core idea of linearly steering reflective states to be straightforward, well-motivated, and interesting. Multiple reviewers (hCwu, zuym, hFui) were concerned that the method relied too heavily on the specific token "Wait" and the Qwen model family. They worried this was a model-specific artifact or a surface-level correlation rather than a generalizable property. Some reviewers (w6Jt and hFui) noted that the presence of reflection tokens and steering vectors are not entirely new concepts, questioning the significance of the contribution.

**Reviewer Concerns:**

In the rebuttal, the authors extended their analysis to more model families beyond Qwen, demonstrating that "wait"-style tokens and the associated steering vectors exist and function similarly across model families. They also analyzed a broader set of reflection keywords to show the vector captures a mode of thinking rather than a single token. Therefore, the concern about the generalization of the finding has been mitigated. However, the technical depth and contribution, as the AC sees, are still marginally below the bar for ICLR, as the method relies on specific reflection keywords to be manually defined instead of automatically learned from data (thus the method appears ad-hoc and sensitive to the specific reflection keywords chosen).

**Reviewer Scores:**

The reviewer scores might slightly increase given that the authors have partly addressed the generalization concern. However, the scores might still not be up to the acceptance bar.

---

### Decision · Program_Chairs · 2026-01-26

Reject